EMBO
Molecular Medicine

# Sodium permeable and "hypersensitive" TREK-1 channels cause ventricular tachycardia

Niels Decher[1],[*],[†] iD, Beatriz Ortiz-Bonnin[1],[†], Corinna Friedrich[3], Marcus Schewe[4] iD, Aytug K Kiper[1], Susanne Rinné[1], Gunnar Seemann[5],[6], Rémi Peyronnet[5],[6], Sven Zumhagen[3], Daniel Bustos[7], Jens Kockskämper[2], Peter Kohl[5],[6], Steffen Just[8], Wendy González[7], Thomas Baukrowitz[4], Birgit Stallmeyer[3] & Eric Schulze-Bahr[3]

## Abstract

In a patient with right ventricular outflow tract (RVOT) tachycardia, we identified a heterozygous point mutation in the selectivity filter of the stretch-activated $K_{2P}$ potassium channel TREK-1 (*KCNK2* or $K_{2P}2.1$). This mutation introduces abnormal sodium permeability to TREK-1. In addition, mutant channels exhibit a hypersensitivity to stretch-activation, suggesting that the selectivity filter is directly involved in stretch-induced activation and desensitization. Increased sodium permeability and stretch-sensitivity of mutant TREK-1 channels may trigger arrhythmias in areas of the heart with high physical strain such as the RVOT. We present a pharmacological strategy to rescue the selectivity defect of the TREK-1 pore. Our findings provide important insights for future studies of $K_{2P}$ channel stretch-activation and the role of TREK-1 in mechano-electrical feedback in the heart.

**Keywords** arrhythmia; $K_{2P}$; RVOT; TREK-1; two-pore domain K+ channel
**Subject Categories** Cardiovascular System; Genetics, Gene Therapy & Genetic Disease

See also: **SAN Goldstein** (April 2017)

## Introduction

Right ventricular outflow tract ventricular tachycardia (RVOT-VT) is a common form of monomorphic ventricular tachycardia (VT) characterized by the absence of structural heart disease and a mostly unknown etiology (Srivathsan *et al*, 2005). Some evidence suggests that RVOT-VT can be triggered by delayed afterdepolarizations (DADs) during β-adrenergic stimulation since a rise in intracellular cAMP will lead to a PKA-dependent phosphorylation of different $Ca^{2+}$ handling proteins, resulting in overloading of intracellular $Ca^{2+}$ in cardiomyocytes. Following the repolarization phase of the cardiac action potential, the $Na^+/Ca^{2+}$ exchanger transports excessive $Ca^{2+}$ out of the cell, producing a transient inward current that generates DADs (Lerman, 2015) which in turn can trigger VT. Despite advances in the understanding of the molecular mechanisms that trigger VT, the genetic basis of RVOT-VT is largely unknown.

Recently, gene mutations in two-pore domain potassium ($K_{2P}$) channels have been discovered as a cause for familial or sporadic forms of migraine (Lafreniere *et al*, 2010) (TRESK), Birk-Barel syndrome (Barel *et al*, 2008) (TASK-3), and a progressive cardiac conduction disorder (Friedrich *et al*, 2014) (TASK-4). These discoveries provide direct proof for the pathophysiological relevance of $K_{2P}$ "leak" channels. The stretch-activated TREK-1 ($K_{2P}2.1$) channel is regulated by a plethora of different physiological stimuli (Feliciangeli *et al*, 2015) and has been proposed to be involved in multiple cellular and pathophysiological processes. In the heart, it has been suggested that TREK-1 channels play a major role in mechano-electrical feedback, since the stretch-activated K+ current (SAK) shortens action potential duration (APD) and decreases the heterogeneity of repolarization, a potentially anti-arrhythmic activity (Kelly *et al*, 2006). Although TREK-1 is one of the most studied cardiac $K_{2P}$ channels, its physiological role in the human heart and cardiac arrhythmias remained elusive.

---

1  Institute of Physiology and Pathophysiology, Vegetative Physiology, Philipps-University of Marburg, Marburg, Germany
2  Institute of Pharmacology and Clinical Pharmacy, Biochemical and Pharmacological Center (BPC), Philipps-University of Marburg, Marburg, Germany
3  Department of Cardiovascular Medicine, Institute for Genetics of Heart Diseases (IfGH), University Hospital Münster, Münster, Germany
4  Institute of Physiology, Christian-Albrechts-University of Kiel, Kiel, Germany
5  Institute for Experimental Cardiovascular Medicine, University Heart Center Freiburg – Bad Krozingen, Medical Center – University of Freiburg, Freiburg, Germany
6  Faculty of Medicine, University of Freiburg, Freiburg, Germany
7  Center for Bioinformatics and Molecular Simulation, University of Talca, Talca, Chile
8  Molecular Cardiology, University Hospital Ulm, Ulm, Germany
   *Corresponding author. Tel: +49 6421 2862148; E-mail: decher@staff.uni-marburg.de
   †These authors contributed equally to this work

# Results

## A heterozygous *KCNK2* (TREK-1) mutation in a patient with RVOT-VT

We systematically investigated the role of $K_{2P}$ channels for inherited forms of cardiac arrhythmias (Friedrich *et al*, 2014). In a large patient cohort comprising 438 probands with different genetically unresolved arrhythmia syndromes [RVOT-VT (40), AFib (10), AVB (16), BrS (188), CPVT (32), iVF (68), PCCD (49), PMVT (13), and SND (22)], all coding exons and adjacent intronic sites of *KCNK2*, encoding the $K_{2P}$ channel TREK-1, were sequenced (Materials and Methods and Appendix Table S1). In one of the RVOT-VT patients (Fig 1A), a heterozygous single nucleotide variant (SNV) (c.800T > C) was identified in exon 5 (Fig 1B), resulting in an amino acid exchange of a highly conserved Ile (p.Ile267Thr) in the selectivity filter (SF, Fig 1B and C) of the second pore domain of TREK-1. In a control cohort of the same ethnicity (*n* = 379), this SNV was absent and consistent with a putative disease-causing mutation, it was only found rarely in the Exome Variant Server (EVS) database (3/13,003 alleles).

Starting at an age of 45 years, the affected patient suffered from recurrent and sudden onset VTs that were triggered by physical exercise. The maximum duration of the VT was > 10 min, but did not result in cardiac syncope or arrest. During sinus rhythm, time intervals measured from a 12-lead surface ECG were normal (Appendix Fig S1). Ischemic heart disease was ruled out by coronary angiography when the patient was 49 years old. Additionally, transthoracic echocardiography and MRI together with contrast imaging were unremarkable for the presence of structural heart disease or cardiomyopathy. Programmed electrophysiological stimulation was unable to induce any supraventricular or VT. Two years later, a fast, broad complex tachycardia with a rate of 240 beats/min was noted during exercise (left bundle branch block (LBBB) with inferior axis), indicating RVOT-VT (Fig 1A). Hereafter, surface ECG showed preterminal negative T-waves in the right precordial leads.

There was no family history of tachycardia or sudden cardiac death, and other family members were not available for clinical or genetic investigations. We therefore performed whole exome sequencing (WES) (Materials and Methods and Appendix Table S2) in this proband to identify or exclude additional relevant variants

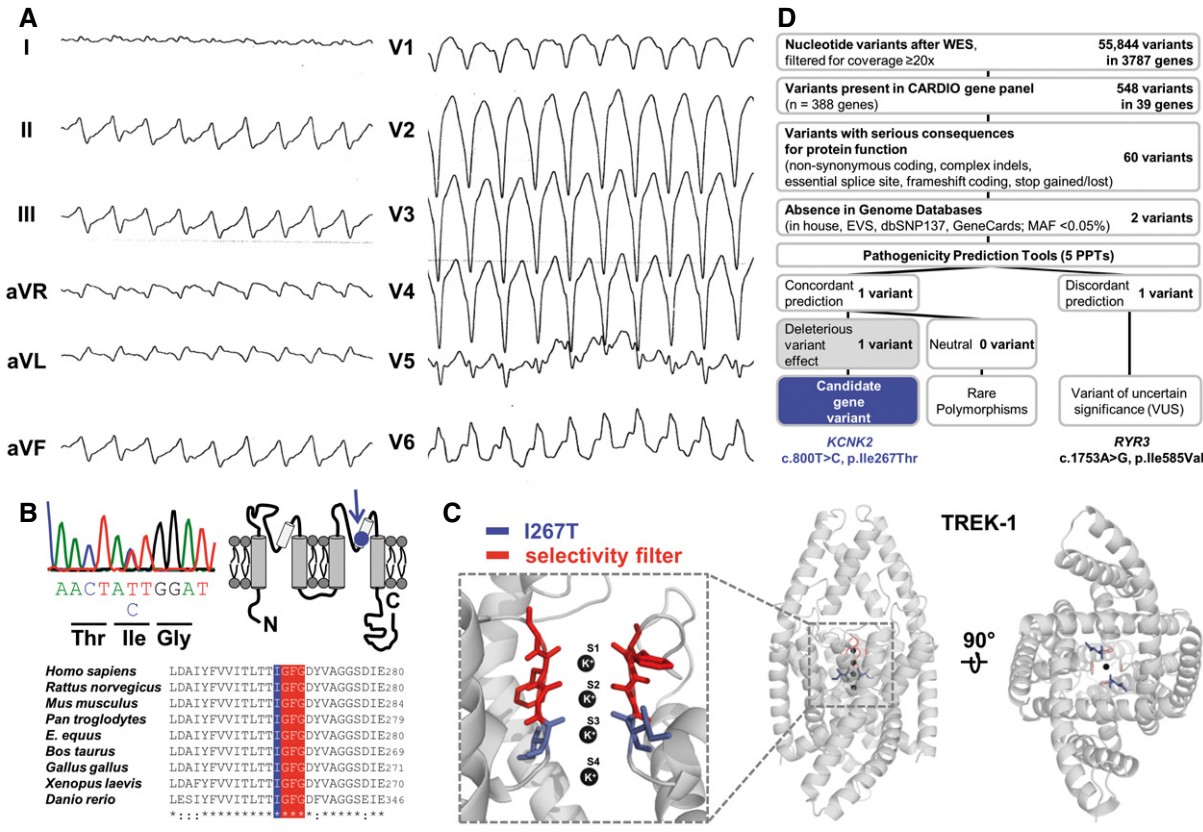

**Figure 1. Identification of a heterozygous *KCNK2* (TREK-1) mutation in a patient with RVOT-VT.**

A    12-lead ECG during exercise of the proband (10772-3) presenting with LBBB (with inferior axis) tachycardia being typical for its origin from the RVOT.

B, C    (B) Electropherogram and nucleotide sequence of *KCNK2* illustrating a heterozygous c.800T > C single nucleotide exchange with a predicted non-synonymous amino acid exchange (p. Ile267Thr; shortly: I276T). Localization of the TREK-1^I267T residue (highlighted in blue) within a highly conserved signature sequence (red) upon partial sequence alignment with TREK-1 orthologues. The I276T mutation is located in the second pore domain of the TREK-1 channel (indicated in a cartoon in B), or in a pore homology model based on the crystal structure of TREK-2 (C).

D    Prioritization scheme for filtering nucleotide variants obtained after WES.

that may be responsible for the arrhythmia phenotype. Following a prioritization scheme (Fig 1D), only SNVs with a minimum sequencing coverage of 20× and that were present in an in-house cardiovascular priority gene list (CARDIO panel; Friedrich *et al*, 2014), harboring 388 relevant cardiovascular or ion channel genes, were further evaluated. Considering only SNVs with potentially serious consequences to the protein, only two were absent or very rare in genomic databases (Appendix Table S3), whereas 58 were already known and not further considered as causative (Appendix Table S4). The two remaining SNVs were independently confirmed by Sanger sequencing and were predicted to cause amino acid exchanges (*KCNK2*: c.800T > C, p.Ile267Thr, and *RYR3*: c.1753A > G, p.Ile585Val). *RYR3* encodes the ryanodine receptor type 3 which is preferentially expressed in the brain and only faintly expressed in the heart. The pathogenic impact of both amino acid exchanges was determined using five different *in silico* pathogenicity prediction tools (PPTs) (Fig 1D and Appendix Table S3). The

identified Ile585Val variant in the *RYR*3 gene had discordant, but mainly non-deleterious pathogenicity predictions (4/5: "neutral") and was not further considered (Fig 1D). Confirming our initial candidate gene approach, WES and the prioritization scheme predicted the I267T exchange in TREK-1 as a likely disease-related variant. All PPTs concordantly evaluated the TREK-1$^{I267T}$ exchange as "damaging" (Appendix Table S3).

## Reduced outward currents of homomeric TREK-1$^{I267T}$ and heteromeric TREK-1/TREK-1$^{I267T}$

Heterologous expression of wild-type TREK-1 channels in *Xenopus* oocytes resulted in large outward currents with a reversal potential ($E_{rev}$) of −90 mV (Fig 2A). In contrast, TREK-1$^{I267T}$ currents were much smaller in magnitude (Fig 2A and B) with an $E_{rev}$ near −50 mV (Fig 2C and D). TREK-1$^{I267T}$ expressing oocytes appeared as leaky cells. However, one can note an unusual "hook" in the

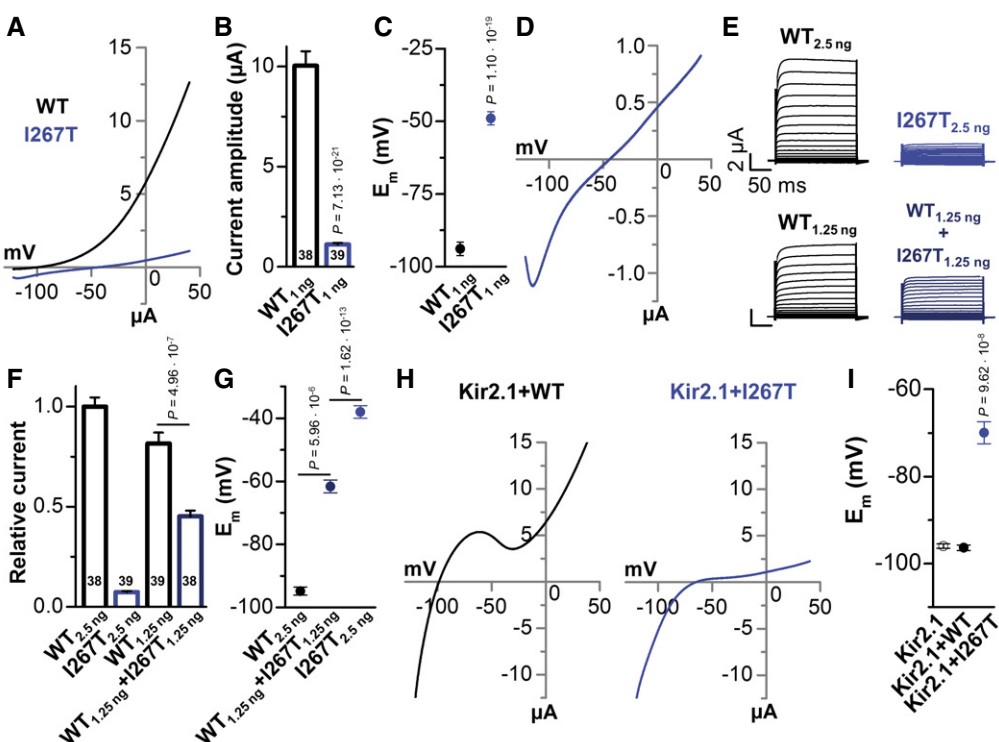

**Figure 2. TREK-1$^{I267T}$ causes a dominant-negative reduction in outward currents and depolarizes membrane potentials.**

A   Representative voltage-clamp measurements in *Xenopus* oocytes injected with wild-type TREK-1 (black) or TREK-1$^{I267T}$ (blue) cRNA (1 ng/oocyte). Currents were recorded 48 h after cRNA injection, applying a ramp protocol, for 3.5 s rising from −120 mV to +40 mV, from a holding potential of −80 mV.
B   Mean current amplitudes of TREK-1- and TREK-1$^{I267T}$-mediated currents analyzed at +40 mV. Numbers of experiments are indicated within the bar graph.
C   Mean membrane potential ($E_m$) of TREK-1 (black) and TREK-1$^{I267T}$ (blue). Numbers of experiments are indicated within the bar graph.
D   Representative measurement of the current–voltage relationship of TREK-1$^{I267T}$.
E   Representative current–voltage relationship measurements in oocytes injected with TREK-1 cRNA (2.5 ng/oocyte), TREK-1$^{I267T}$ (2.5 ng/oocyte), wild-type 1.25 ng TREK-1 (mimicking haploinsufficiency), or with wild-type TREK-1 and TREK-1$^{I267T}$ cRNA (1.25 ng each/oocyte, mimicking a heterozygous state). Current–voltage relationship was recorded from a holding potential of −80 mV, applying 10 mV steps from −130 to +50 mV with a duration of 200 ms.
F   Analysis of the relative current amplitude of (E) at +40 mV normalized to TREK-1 wild-type currents (2.5 ng). Numbers of experiments are indicated within the bar graph.
G   Analysis of the $E_m$ of (E and F).
H   Representative current traces of Kir2.1 co-expressed with TREK-1(WT) or the I267T mutant.
I   $E_m$ analysis from (H) (n = 13) and of Kir2.1 expressed alone.

Data information: Data are presented as mean ± SEM. Data in (B, G, and I) are analyzed by non-parametric Mann–Whitney *U*-test. Data in (C and F) are analyzed by unpaired Student's *t*-test. *P*-values are indicated.

    

most negative voltage range of the current–voltage relationship (Fig 2D). This unusual feature that was not observed in uninjected oocytes, results from the rate of activation of mutant channels after the voltage is stepped to −120 mV (Appendix Fig S2). Co-injection of oocytes with equal amounts of wild-type TREK-1 plus TREK-1$^{I267T}$ cRNA (to mimic the heterozygous state) results in reduced current compared to that observed in oocytes injected with an equal amount of wild-type TREK-1 cRNA alone (Fig 2E and F). Therefore, the mutation might act in a dominant-negative manner. Intriguingly, also for the co-expression, mimicking heterozygosity, the membrane potential ($E_m$) was strongly depolarized (Fig 2G). Generally, depolarized membrane potentials could result from reduced outward $K^+$ currents or altered ion selectivity. In ventricular myocytes, the resting membrane potential is primarily determined by Kir2.x channels. Kir2.1 co-expressed with TREK-1 showed typical Kir2.1-like currents (Fig 2H), while co-expression with TREK-1$^{I267T}$ results in a dramatic depolarization of the $E_{rev}$ and the $E_m$ (Fig 2H and I). Blocking TREK-1 channels under these conditions only mildly depolarizes cells ($E_m$ is already set by Kir2.1), while it hyperpolarizes cells expressing TREK-1$^{I267T}$ (Appendix Fig S3), indicating that the TREK-1 mutation has the inherent potential to depolarize cells.

## Na$^+$ permeability in TREK-1$^{I267T}$

The membrane depolarization associated with expression of TREK-1$^{I267T}$ mutant channels might be achieved via an altered ion selectivity. Thus, the relative permeability of mutant channels was assessed by gradually replacing extracellular $Na^+$ by $K^+$ (Fig 3A and B). While for TREK-1, a linear relationship between the extracellular $K^+$ and the $E_{rev}$ was observed, reflecting the Nernst equation, the TREK-1$^{I267T}$ mutant showed a loss of $K^+$ selectivity (13.2 mV/decade) (Fig 3B). In contrast, when extracellular $Na^+$ was gradually replaced by NMDG$^+$, the cells expressing homomeric or heteromeric TREK-1$^{I267T}$ hyperpolarized near to the $K^+$ equilibrium potential (Fig 3C), indicating that the mutant channel is permeable for $Na^+$.

TREK-1 is "activated" by extracellular sodium (Fink *et al*, 1996), accordingly, replacing extracellular $Na^+$ by NMDG$^+$ strongly reduced wild-type TREK-1 outward currents (Fig 3D). In contrast, for TREK-1$^{I267T}$ the outward currents were increased, the leak with its unique current kinetics was removed and the $E_{rev}$ was hyperpolarized to a value close to $E_K$ (Fig 3D). Thus, extracellular $Na^+$ activates the outward conductance of wild-type TREK-1, but causes

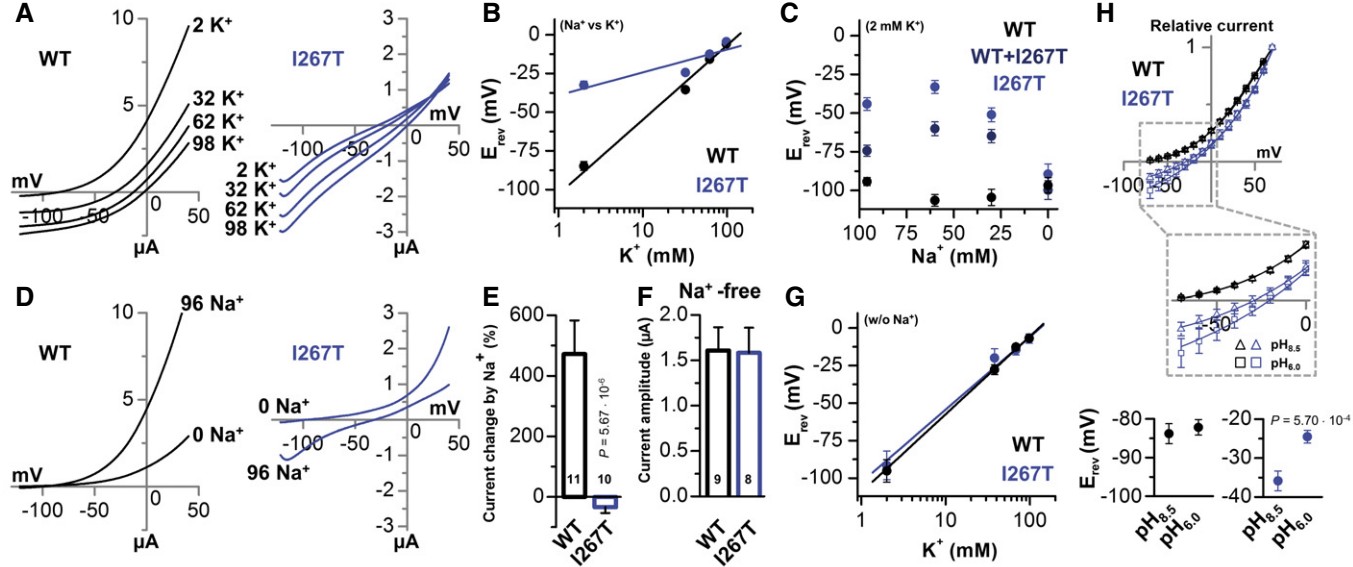

**Figure 3. The TREK-1$^{I267T}$ mutation alters the potassium selectivity and introduces a sodium permeability.**

A, B  (A) Representative voltage-clamp recordings of wild-type TREK-1 (black) and TREK-1$^{I267T}$ (blue) using different extracellular $K^+$ concentrations, by replacing extracellular NaCl to KCl, and (B) the respective analysis of the $E_{rev}$ of TREK-1 (black) or TREK-1$^{I267T}$ (blue) as a function of the external $K^+$ concentration ($n = 10$). The slope of the TREK-1$^{I267T}$ mutant was only 13.2 mV/decade.

C  Analysis of the $E_{rev}$ of TREK-1 (black), mutant TREK-1$^{I267T}$ (blue), and TREK-1 co-expressed with TREK-1$^{I267T}$ (dark blue) while gradually removing the extracellular $Na^+$, by replacing NaCl to NMDG-Cl ($n = 12$). The $K^+$ concentration was maintained at 2 mM.

D  Representative current traces of TREK-1 (black) and TREK-1$^{I267T}$ (blue) recorded in a bath solution containing 96 mM $Na^+$ or 0 $Na^+$ (NaCl to NMDG-Cl with K$_o^+$ maintained at 2 mM).

E  Percentage of current change when $Na^+$ was removed from the extracellular recording solution, as in (D). Numbers of experiments are indicated within the bar graph.

F  Average current amplitude of TREK-1 or TREK-1$^{I267T}$ in extracellular solution containing 98 mM $K^+$ but no $Na^+$ (all NaCl replaced to KCl). Numbers of experiments are indicated within the bar graph.

G  $E_{rev}$ of wild-type TREK-1 (black) and TREK-1$^{I267T}$ (blue) as a function of the external $K^+$ concentration (w/o $Na^+$, 0 mM $Na^+$) ($n = 9$). The slope was about 52 mV/decade for both constructs.

H  Normalized current–voltage relationship of TREK-1 (black) ($n = 7$) and TREK-1$^{I267T}$ (blue) ($n = 6$) at pH$_o$ 8.5 and 6.0. The inset shows a magnification, highlighting the different $E_{rev}$. Panels at the bottom illustrate the average of the $E_{rev}$ at pH 8.5 and 6.0 of TREK-1 (black) and TREK-1$^{I267T}$ (blue).

Data information: Data are presented as mean ± SEM. Data in (E) are analyzed by non-parametric Mann–Whitney U-test. Data in (H) are analyzed by unpaired Student's t-test. P-values are indicated.

    

Na$^+$ inward currents and a partial reduction in the outward currents recorded for the TREK-1$^{I267T}$ mutant (Fig 3D and E). In the absence of Na$^+$, wild-type and mutant TREK-1 channels conduct outward currents with similar amplitudes (Fig 3F), indicating that the reduced outward currents of TREK-1$^{I267T}$ are caused by their greatly enhanced Na$^+$ permeability, while the surface expression of the mutant is most likely not affected. As expected, determining the selectivity in Na$^+$-free solution (NMDG$^+$ by K$^+$) the TREK-1$^{I267T}$ mutant obeys the Nernst equation for a selective K$^+$ channel (Fig 3G). In addition, we have performed molecular dynamic simulations which are also in agreement with the loss of selectivity effect of TREK-1$^{I267T}$ (Appendix Fig S4).

### TREK-1$^{I267T}$ has a TWIK-like sodium permeable selectivity filter

TREK-1 has two pore loops containing the amino acid sequence "TIGFG". In TWIK-1, the pore signature sequence of the first pore loop differs from that of other K$_{2P}$ channels. Here, the conserved isoleucine preceding the first signature sequence is substituted by a threonine ("TTGYG"; Ma *et al*, 2011; Chatelain *et al*, 2012), which

results in an enhanced sodium permeability of these channels under hypokalemic (Ma *et al*, 2011) or acidic conditions (Chatelain *et al*, 2012; Ma *et al*, 2012). The TREK-1$^{I267T}$ mutation in the second pore loop creates a TWIK-like selectivity filter, potentially explaining the high Na$^+$ ion permeability observed in the mutant channel which, however, already occurs under baseline conditions. Similar to TWIK-1 (Chatelain *et al*, 2012; Ma *et al*, 2012) and unlike wild-type TREK-1, the E$_{rev}$ of TREK-1$^{I267T}$ expressing cells is depolarized upon extracellular acidification (Fig 3H and Appendix Fig S5).

### TREK-1$^{I267T}$ depolarizes HL-1 cardiomyocytes and slows upstroke velocity

Consistent with the overexpression of a K$^+$ channel, transfection of wild-type TREK-1 (Fig 4A) slowed the spontaneous beating frequency of HL-1 cardiomyocytes (Fig 4B). In contrast, the action potential (AP) frequency of HL-1 cells overexpressing TREK-1$^{I267T}$ was significantly increased compared to wild-type transfected cells (Fig 4B). Patch clamp recordings showed that the overexpression of TREK-1$^{I267T}$ did not significantly alter the AP repolarization

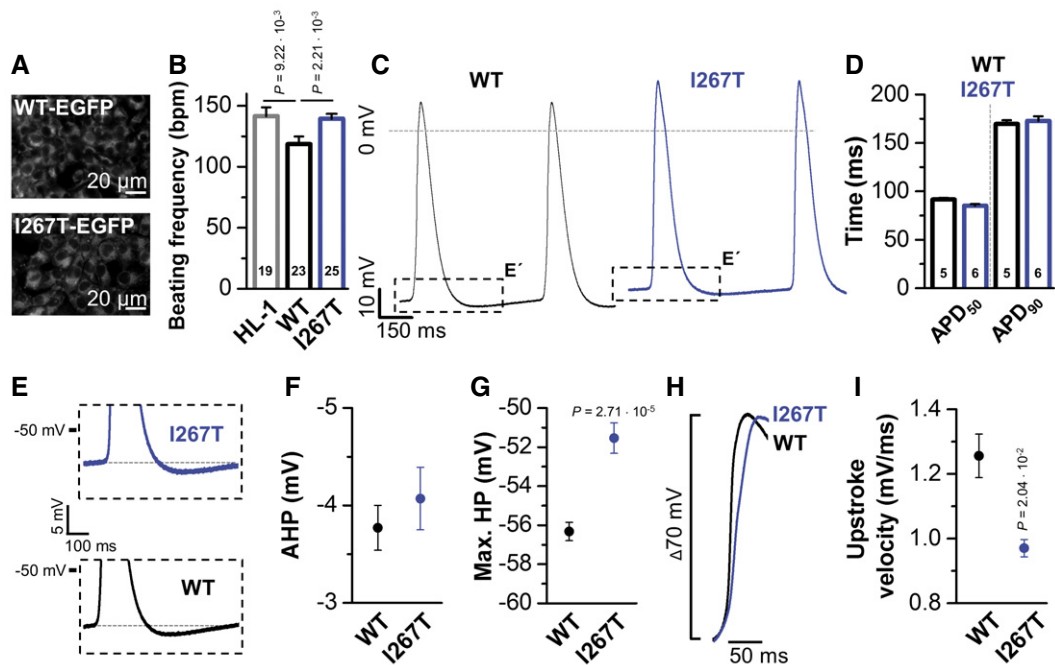

**Figure 4. TREK-1$^{I267T}$ speeds early repolarization and depolarizes the membrane potential of spontaneously beating HL-1 cells.**

A    Fluorescence imaging of HL-1 cells transfected with EGFP-tagged TREK-1 or TREK-1$^{I267T}$.

B    Action potential (AP) frequency, as beats per minute (bpm), counted in HL-1 cells (HL-1) bathed in Claycomb medium containing 100 μM norepinephrine or HL-1 cells transfected with TREK-1-EGFP (WT) or TREK-1$^{I267T}$-EGFP (I267T). The number of experiments are indicated in the bar graphs.

C    Patch clamp measurements in the current-clamp mode of HL-1 cells transfected with TREK-1-EGFP or mutant TREK-1$^{I267T}$-EGFP. Boxes indicate the zoom area for panel (E).

D    Analyses of the AP duration, APD$_{50}$, and APD$_{90}$ of HL-1 cells transfected with TREK-1-EGFP or TREK-1$^{I267T}$-EGFP. The number of experiments are indicated in the bar graphs.

E    Illustration of the hyperpolarization observed following an AP of HL-1 cells transfected with TREK-1-EGFP or TREK-1$^{I267T}$-EGFP.

F    Analysis of the afterhyperpolarization (AHP) observed in HL-1 cells transfected with TREK-1-EGFP (n = 5) or TREK-1$^{I267T}$-EGFP (n = 6).

G    Analyses of the maximum diastolic hyperpolarization (max. HP) of HL-1 cells transfected with TREK-1-EGFP (n = 5) or TREK-1$^{I267T}$-EGFP (n = 6).

H    Illustration of the upstroke velocity of HL-1 cells transfected with TREK-1-EGFP or TREK-1$^{I267T}$-EGFP. The threshold for the action potentials was aligned in order to compare the upstroke phase and velocity.

I    Analyses of the upstroke velocity (mV/ms) of HL-1 cells transfected with TREK-1-EGFP (n = 5) or TREK-1$^{I267T}$-EGFP (n = 6).

Data information: Data are presented as mean ± SEM. Data in (B, G, and I) are analyzed by non-parametric Mann–Whitney *U*-test. *P*-values are indicated.

(Fig 4C and D). The shape of the afterhyperpolarization (AHP) was not altered (Fig 4E and F). As expected for a $Na^+$ permeable channel, HL-1 cells expression TREK-1$^{I267T}$ were more depolarized (Fig 4C, E and G), an effect that would be pro-arrhythmic in ventricular cardiomyocytes, as the membrane potential is closer to the AP threshold. In addition, a slowing of the upstroke velocity was observed (Fig 4H and I), a factor that is known to contribute to ventricular re-entry arrhythmias due to conduction velocity reduction and thus a shorter wavelength. Next we utilized a computational model of the human ventricular action potential (Appendix Supplementary Methods and Appendix Fig S6) and assigned a background potassium current with a relative sodium permeability of 20%, as observed for TREK-1$^{I267T}$. Consistent with our experimental observations, a mild depolarization and a slowing of the upstroke velocity were present (Appendix Fig S6H and I).

### BL-1249 restores TREK-1$^{I267T}$ $K^+$ selectivity

Based on our observation that inhibition of TREK-1 in cells co-expressing TREK-1$^{I267T}$ and Kir2.1 results in hyperpolarization of cells (Appendix Fig S3), blocking TREK-1$^{I267T}$-mediated currents may be a suitable approach for arrhythmia prevention in affected individuals. In this context, we found that verapamil, which is used to treat the patient, is a blocker of TREK-1 channels, exhibiting a similar affinity as fluoxetine (Kennard *et al*, 2005; Appendix Fig S7). However, Kv channel blockers often interact with the residue equivalent to I267 in TREK-1 (Decher *et al*, 2004). Indeed, TREK-1$^{I267T}$ channels have a reduced sensitivity to block by fluoxetine or verapamil (Appendix Fig S7), which is likely to prevent the usage of these drugs to block this pathological $Na^+$ leak.

Next, oocytes expressing TREK-1$^{I267T}$ were incubated with different TREK-1 blockers and activators to probe for drugs that "rescue" the ion selectivity defect. Verapamil and fluoxetine failed to restore $K^+$ selectivity (Fig 5A). Similarly, the TREK-1 channel activators, 2-APB (Beltran *et al*, 2013) and riluzole (Duprat *et al*, 2000), had a reduced affinity and did not rescue I267T selectivity (Fig 5A) and instead caused an additional depolarization (Appendix Fig S8A and B). However, BL-1249 (Veale *et al*, 2014) fully restored TREK-1 characteristics (Fig 5A, B, E and F, and Appendix Figs S8C and S9) and hyperpolarized membrane potentials. BL-1249 activates the TREK-1$^{I267T}$ mutant with similar affinity (Fig 5C and D), indicating a different binding site of BL-1249 compared to the other compounds studied. Thus, BL-1249 seems to indirectly stabilize the SF as suggested by restoration of the $K^+$ selectivity of TREK-1$^{I267T}$ (Fig 5E and F).

### TREK-1$^{I267T}$ channels are hypersensitive to mechanical stretch

A hallmark of the TREK/TRAAK $K_{2P}$ channel subgroup is their sensitivity to mechanical stretch as assessed by applying negative pressure via the pipette to inside-out patches (Fig 6). This stretch-sensitivity might be of major relevance for mechano-electrical feedback in the heart and possibly the pathogenesis of ventricular arrhythmias. We found that TREK-1$^{I267T}$ channels displayed a markedly increased stretch-sensitivity. Higher peak currents were evoked with less negative pressure, when normalized to the peak current induced by $pH_i$ 5.0 activation (Fig 6A and B, and Appendix Fig S10). Furthermore, the characteristic desensitization of the stretch-induced currents in TREK-1 was slowed by the I267T mutation. The time constant for the fast component of desensitization was slowed

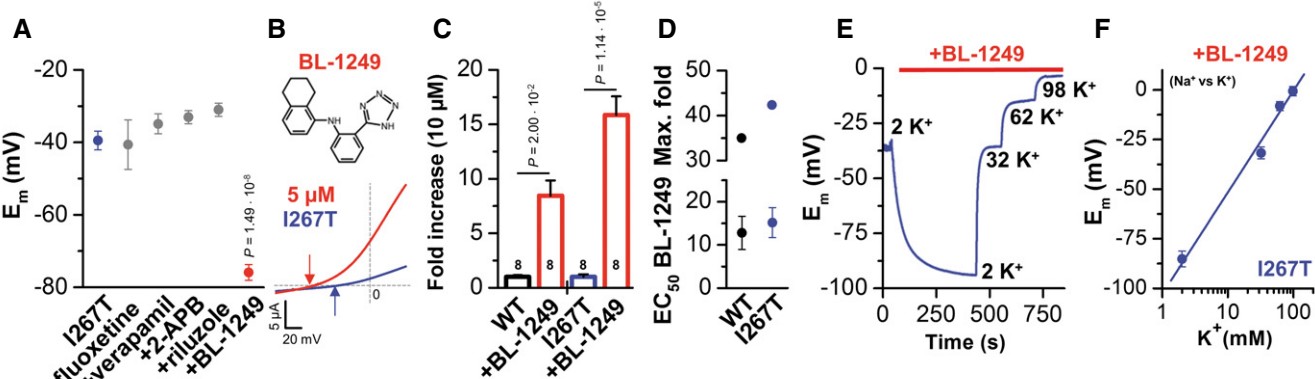

**Figure 5. TREK-1$^{I267T}$ reduces the affinity of TREK-1 channel blockers and activators, but TREK-1$^{I267T}$ is rescued by the activator BL-1249.**

A   Analysis of the $E_m$ of TREK-1$^{I267T}$ channel-mediated currents without drug application (*n* = 15, blue) and in the presence of different drugs. 80 μM fluoxetine (*n* = 8), 62 μM verapamil (*n* = 13), 50 μM 2-APB (*n* = 4), 500 μM riluzole (*n* = 4), or 5 μM BL-1249 (*n* = 8, red) were tested.

B   Chemical structure of BL-1249. Representative current trace of the TREK-1$^{I267T}$ mutant recorded in ND96 solution (blue) and after application of 5 μM BL-1249 (red). The arrows indicate the respective reversal potential.

C   Fold increase in TREK-1 and TREK-1$^{I267T}$ channel-mediated currents by 10 μM BL-1249. Numbers of experiments are indicated within the bar graph.

D   The lower panel shows the $EC_{50}$ of BL-1249. Native TREK-1 $EC_{50}$ = 12.8 ± 3.8 μM (*n* = 3); TREK-1$^{I267T}$ $EC_{50}$ = 15.2 ± 3.4 μM (*n* = 8). The upper panel shows the maximum fold increase derived from the Hill equation; TREK-1: 35-fold, TREK-1$^{I267T}$: 42.4-fold.

E   $E_m$ of TREK-1$^{I267T}$ expressing oocytes depending on BL-1249 and on the extracellular $K^+$ concentration in the presence of BL-1249.

F   The $E_m$ of the TREK-1$^{I267T}$ mutant in the presence of BL-1249 was plotted against the extracellular $K^+$ concentration. The slope was 51.2 mV/decade (*n* = 6).

Data information: Data are presented as mean ± SEM. Data in (A) are analyzed by non-parametric Mann–Whitney *U*-test. Data in (C) are analyzed by non-parametric Mood's median test. *P*-values are indicated.

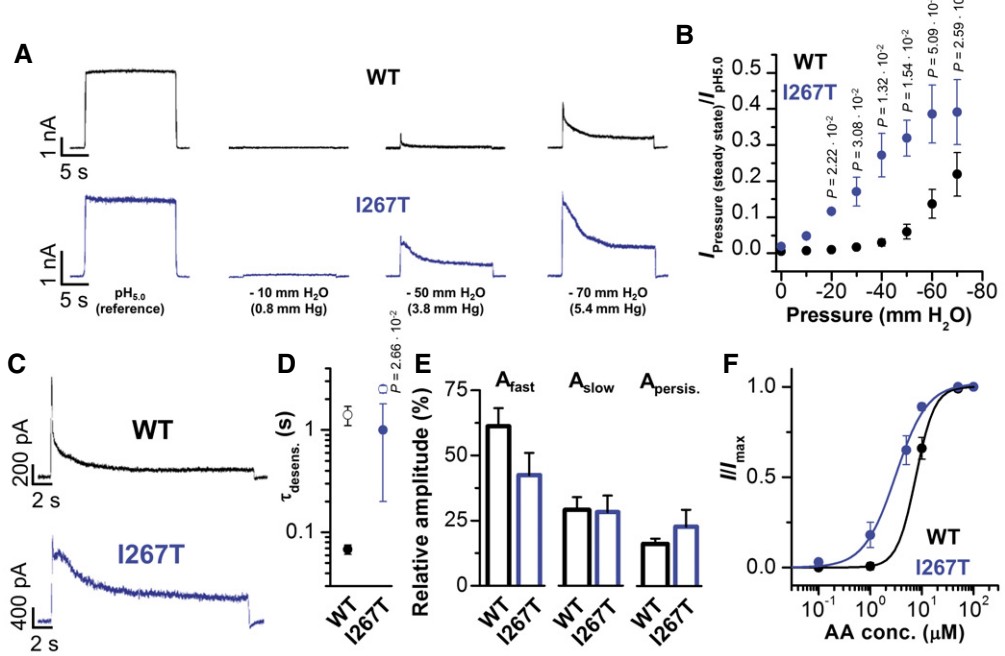

**Figure 6. TREK-1$^{I267T}$ has an increased stretch-sensitivity, a slowed stretch-desensitization, and increased arachidonic acid sensitivity.**

A  Representative inside-out macropatch measurements of TREK-1 (black) and TREK-1$^{I267T}$ (blue) at different negative pressures applied via the pipette.
B  Normalized current at the steady state of desensitization plotted against the negative pressure applied for TREK-1 (black) and TREK-1$^{I267T}$ (blue) ($n = 5$).
C  Representative macropatch recordings at $-50$ mm $H_2O$ column of TREK-1 (black) and TREK-1$^{I267T}$ (blue).
D  Average time constants of desensitization ($\tau_{desens.}$) derived from the bi-exponential fit of current traces at $-50$ mm $H_2O$ column; $\tau_{fast}$ (filled circles) and $\tau_{slow}$ (open circles) ($n = 5$).
E  Relative amplitudes of the fast and the slow component of current desensitization and the non-desensitizing persistent current for TREK-1 and TREK-1$^{I267T}$ ($n = 5$).
F  Relative current activation of TREK-1 (black) or TREK-1$^{I267T}$ (blue) using different intracellular arachidonic acid (AA) concentrations. TREK-1 ($n = 7$); TREK-1$^{I267T}$ ($n = 8$).

Data information: Data are presented as mean $\pm$ SEM. Data in (B) are analyzed either by unpaired Student's $t$-test or by Welch's $t$-test. Data in (D) are analyzed by unpaired Student's $t$-test. $P$-values are indicated.

more than 10-fold (from $68 \pm 7$ ms to $1.0 \pm 0.8$ s) and the slow component about twofold (from $1.4 \pm 0.3$ s to $2.5 \pm 0.2$ s) (Fig 6C–E). TREK-1$^{I267T}$ currents exhibited a reduced fast component of desensitization (Fig 6C and E) and due to the slowing of stretch-desensitization, a tendency to increased persistent currents (Fig 6E). Activation of TREK-1 by arachidonic acid (AA) is proposed to occur by a process linked to stretch-activation (Brohawn, 2015). The EC$_{50}$ for AA activation decreased from $6.9 \pm 1.2$ μM (WT) to $3.6 \pm 0.6$ μM (TREK-1$^{I267T}$) (Fig 6F and Appendix Fig S11), demonstrating a mechanistic link between stretch- and lipid-activation of these channels (Brohawn, 2015).

**Altered β$_1$-adrenoreceptor modulation in TREK-1$^{I267T}$ channels**

Attacks of RVOT-VTs arise under β-adrenergic stimulation; therefore, we tested how β$_1$-adrenergic stimulation influences wild-type and TREK-1$^{I267T}$ channels co-expressed with the β$_1$-adrenoreceptor in oocytes. Strikingly, while wild-type TREK-1 currents are almost fully suppressed by β$_1$-adrenoreceptor stimulation (Fig 7), the TREK-1$^{I267T}$ mutant is not responding in the typical manner (Fig 7A–C), the aberrant Na$^+$ conductivity persists (Fig 7B), and most importantly, there are even more pronounced inward leak currents (Fig 7D and E) with an additional depolarization of the

cells (Fig 7B). Thus, under sympathetic stimulation, the depolarizing impact of the mutant is more pronounced as under baseline conditions–an effect that might in our particular patient contribute to episodes of VTs under stress conditions.

## Discussion

The lipid- and stretch-activated K$_{2P}$ channel TREK-1 has been proposed to play an important role in cardiac mechano-electrical feedback. However, the physiological relevance of this channel for the human heart and for cardiac arrhythmias remained elusive. Moreover, a monogenetic basis for RVOT-VT has not been established so far. Despite the lack of a genetically informative family, the whole exome analysis in the index patient, together with the comprehensive functional data revealing an increased Na$^+$ permeability and stretch-activation, strongly supports that mutant TREK-1 channels cause RVOT-VT. Despite the fact that TREK-1 is also expressed in the brain and the adrenal gland, extra-cardiac phenotypes [e.g., a neurological disorder or primary hyperaldosteronism caused by a Na$^+$ leak in the adrenal gland, as observed for somatic mutations in *KCNJ5* (Gomez-Sanchez & Oki, 2014; Kuppusamy et al, 2014)] were not observed in the patient. As the heteromeric

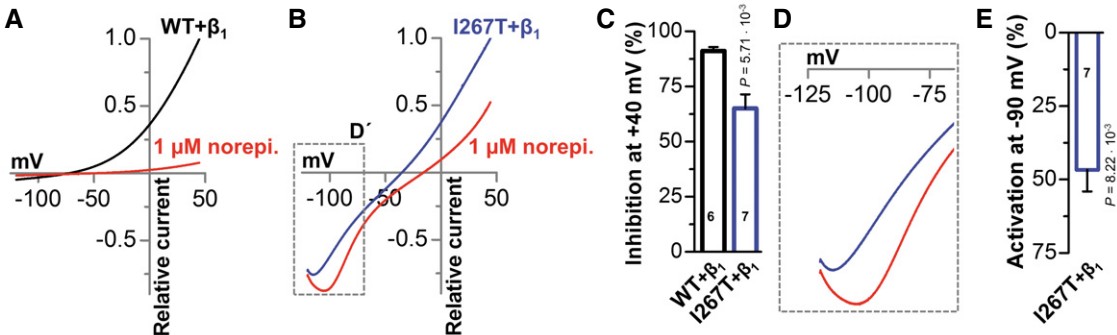

**Figure 7. Altered $\beta_1$-adrenoreceptor modulation in TREK-1$^{I267T}$ channels.**

A Voltage-clamp measurement of wild-type TREK-1 co-expressed with a $\beta_1$-adrenoreceptor in *Xenopus* oocytes before (black) and after (red) application of 1 $\mu$M norepinephrine. Currents were recorded from a holding potential of $-80$ mV applying a ramp protocol for 3.5 s rising from $-120$ mV to $+40$ mV. Illustrated are the currents averaged from six cells.

B Voltage-clamp measurement of TREK-1$^{I267T}$ co-expressed with a $\beta_1$-adrenoreceptor in *Xenopus* oocytes before (black) and after (red) application of 1 $\mu$M norepinephrine. Illustrated are the currents averaged from seven cells.

C Analysis of the $\beta_1$-adrenoreceptor-mediated inhibition of wild-type TREK-1 (black) and TREK-1$^{I267T}$ (blue) at $+40$ mV by 1 $\mu$M norepinephrine. The number of experiments is indicated within the respective bars.

D Zoom in from (B) to hyperpolarized potentials, illustrating the increased inward currents of TREK-1$^{I267T}$ after application of 1 $\mu$M norepinephrine (red).

E Quantification of the activation of inward currents by TREK-1$^{I267T}$ at $-90$ mV by 1 $\mu$M norepinephrine. The number of experiments is indicated within the bar.

Data information: Data are presented as mean $\pm$ SEM. Data in (C) are analyzed by Welch's *t*-test. Data in (E) are analyzed by paired Student's *t*-test. *P*-values are indicated.

channels of TREK-1$^{I267T}$ with wild-type TREK-2 are not Na$^+$ permeable (Appendix Fig S12), the clinical phenotype might depend on a tissue-dependent heteromerization of TREK-1 with TREK-2. Consistent with the ventricular phenotype in our patient, relatively low expression of TREK-2 was found in the heart and its expression was mainly limited to the atria (Gu *et al*, 2002; Liu & Saint, 2004). As the TREK-2 expression is also relatively low in the adrenal cortex (Choi *et al*, 2011), other mechanisms must contribute to the lack of an adrenal phenotype, i.e., heteromerization with other K$_{2P}$ channels expressed in the adrenal cortex like TWIK-1 (Choi *et al*, 2011; Hwang *et al*, 2014) or dominant-negative splice variants as TREK-1e (Rinné *et al*, 2014).

Noteworthy, alternative translation initiation (ATI) can result in a TREK-1 variant with a truncated N-terminus (TREK-1$\Delta$N) that is permeable to sodium (Thomas *et al*, 2008). While a recent study (Veale *et al*, 2014) suggested that TREK-1$\Delta$N is highly potassium selective, the authors only studied the channels in the presence of fenamates (including BL-1249) to increase current density, the same agents that we show here to restore potassium selectivity in sodium permeable TREK-1$^{I267T}$ channels (Fig 5).

Our analysis of the I267T mutant also provides valuable insights into the gating of stretch-activated K$_{2P}$ channels such as TREK and TRAAK which can be activated by mechanical force directly through the lipid bilayer (Brohawn *et al*, 2014; Brohawn, 2015). The coupling of force to gating involves structural changes in the protein that are thermodynamically favorable, including "protein expansion" and changes in "lipid deformation" (Brohawn *et al*, 2014; Brohawn, 2015). These changes can be induced by membrane tension and involve: (i) A protein expansion in the inner leaflet due to conformational changes in both TM4 and TM2-TM3, specifically when channels transition into the so-called "up" state, which is thought to represent the open state of the channel (Brohawn, 2015). (ii) Membrane tension favors the "up" state, as rearrangement of these

TM helices creates a flatter membrane facing surface (cylindrical instead of wedge shaped) (Brohawn, 2015), which is favorable as it reduces the membrane curvature or midplane bending deformation (Phillips *et al*, 2009; Brohawn, 2015). Brohawn (2015) proposed that PUFAs like AA might act in this context by lowering the lipid deformability barrier, so that already in the absence of membrane tension the energetically favorable "up" or open state is promoted. The increased stretch-sensitivity of mutant TREK-1 channels which we have observed was paralleled by an increased lipid sensitivity, providing experimental evidence that these two effects are, as previously proposed (Brohawn, 2015), mechanistically tightly coupled. Moreover, our results strongly suggest that mechano-activation involves structural changes in the SF because the I267T mutation clearly affects this structure, as evident by the altered ion selectivity. Interestingly, it was recently revealed that ion pore occupancy is a major determinant in the gating of K$_{2P}$ channels, including TREK-1 (Schewe *et al*, 2016). The mechanism of stretch-desensitization had previously been enigmatic. Here we provide the first experimental evidence that this process is directly linked to structures of the SF.

It has been proposed that VTs that originate from the RVOT may have their origin during development of the heart (Boukens *et al*, 2016). Perhaps the sodium permeability observed for TREK-1$^{I267T}$ causes subtle changes in the embryonic development of the RVOT which only become evident during adult life, as proposed by Boukens *et al* (2016). Specifically, these authors proposed that arrhythmia risk requires multiple factors, for example, an ion channel mutation together with increased wall stress. In this context, it is noteworthy that occlusion of the outflow tracts has been reported to cause ventricular extrasystoles that apparently generate from the occlusion site (Franz, 1996).

As mentioned above, membrane stretch can induce premature ventricular excitations (Franz *et al*, 1992; Stacy *et al*, 1992) or runs of VT (Hansen *et al*, 1990; Kelly *et al*, 2006). The underlying mechanism

is probably depolarization via stretch-activated nonselective cation currents (SAC). In contrast, stretch-activated $K^+$ currents (SAK), which are thought to be conducted by cardiac TREK-1 channels, cause AP shortening and stabilization of the membrane potential (Kelly *et al*, 2006; Xian Tao *et al*, 2006). However, the regional density of SAC and SAK in the heart is unknown and quantitative analysis of expression levels is complicated by the fact that the molecular identity of the channel(s) conducting the SAC component has not been defined. In addition, there are differences of mechanical strain across the ventricular wall and in different regions of the heart (Kelly *et al*, 2006). Clearly, an important role of SAKs may be in controlling the regional or transmural dispersion of repolarization in the heart or across the ventricular wall (Kelly *et al*, 2006), an effect that will be anti-arrhythmic by an improved synchronization of repolarization (Kelly *et al*, 2006). A loss of SAK components, as in the knock-out of the TREK-1, TREK-2, and TRAAK genes, induces a hypersensitivity to mechanical force, a mechanical allodynia, and an enhanced mechanical hyperalgesia, and the severity of the phenotype is increased in double or triple knock-out mice (Heurteaux *et al*, 2004; Alloui *et al*, 2006; Noel *et al*, 2009). In the heart, TREK-1$^{I267T}$ mutant channels would not only cause a loss of the membrane-stabilizing SAK current, it would also generate a new stretch-activated SAC conductance permeable to $Na^+$. Thus, mutant TREK-1$^{I267T}$ channels might cause intracellular $Na^+$ overload and depolarization of cardiac cells, effects which could trigger arrhythmias via secondary $Ca^{2+}$ overload. This effect can be aggravated by the fact that the I267T mutant generates a $Na^+$ permeable channel with increased stretch-sensitivity. The combination of these effects might trigger arrhythmias in cardiac regions that are prone to physical strain, like the RVOT.

# Materials and Methods

### Ethical approval

The study was in accordance with the ethical standards of the Declaration of Helsinki in its latest, revised version and the NIH Belmont Report. Experiments were approved by the local institutional review boards, including the ethic committee of the Ärztekammer Westfalen-Lippe and the Faculty of Medicine of the Westfälische Wilhelms-University. Written informed consent was obtained from all individuals and patients, prior to the study.

### Candidate gene approach

Genomic DNA of all participants in the study was extracted from whole blood according to standard procedures. In a candidate gene approach for RVOT-VT, coding regions and exon-intron boundaries of the *KCNK2* gene were completely sequenced in 40 patients with RVOT-VT as described earlier (Schulze-Bahr *et al*, 2003). The NCBI accession number of *KCNK2* reported in this paper is NM_001017425.2. In one patient (PID: 10772-3), a single nucleotide exchange (c.800T > C) was identified and results in a non-synonymous amino acid exchange (p.Ile267Thr; shortly: I267T). Exon 5 of *KCNK2*, including the identified nucleotide exchange, was sequenced in a control cohort of 379 unrelated, healthy individuals. Primer sequences and PCR conditions for *KCNK2* analysis are available on request.

### Subject ascertainment and phenotypic analysis

A total of 438 patients with various inherited arrhythmia syndromes were analyzed in this study and finally completely sequenced for the presence of a *KCNK2* gene mutation. Comprehensive phenotypic analyses in the patient populations included history, physical examination, and electrocardiography (ECG). In addition, in some patients, coronary angiography and cardiac magnetic resonance imaging were performed according to diagnosis criterion (Priori *et al*, 2013). The control cohort comprised 379 unrelated, healthy Caucasians.

### Whole exome sequencing (WES) in a patient (10772-3) with RVOT-VT

Whole exome sequencing was performed according to standard protocols. Enrichment, library preparation, sequencing, read mapping, and alignment were performed as described previously (Friedrich *et al*, 2014). A total of 4.98 Gbp of sequence data were read and 99.06% were aligned to the reference genome (hg19 build) (Appendix Table S2). The median read depth was 63.6×, and 88.14% of the on-target regions were covered by a depth of at least 20×. Subsequently, variant filtering in these regions was considered to ensure a good detection sensitivity; only 0.27% of the exome sequences were not covered (Appendix Table S2).

### Nucleotide variant annotation and analysis

Nucleotide variants were prioritized according to a scheme published recently (Friedrich *et al*, 2014) to select variants and identify candidate genes for RVOT-VT. In brief, only variants within reliable analyzed genomic regions (sequence coverage > 20×) were included for further analysis. Next, variants within 388 prioritized cardiovascular genes known to be associated with a relevant or inherited cardiac function (CARDIO gene panel; Friedrich *et al*, 2014) were included, and variants in other genes separated. Within this selected gene panel group, all non-genic, intronic, and synonymous variants were excluded and only alterations with a potentially serious consequence (namely amino acid exchanges due to non-synonymous coding, essential splice site, frameshift coding, stop gained or stop lost, or complex indel variation) were further evaluated. Hereafter, all variants being present in the EVS at the NHLBI (EVS, http://evs.gs.washington.edu/EVS/), dbSNP 137 (www.ncbi.nlm.nih.gov/projects/SNP/), or Ensembl Gene Browser (http://www.ensembl.org) (MAF ≥ 0.05%) were excluded due to the potential of being a rare polymorphism and remaining variants were finally confirmed by direct bi-directional Sanger sequencing in two independent reactions.

### Pathogenicity prediction tools

The pathogenic impact of validated amino acid substitutions was predicted by PolyPhen-2, MutPred, SNAP, SNPs&GO, and SIFT as previously described (Friedrich *et al*, 2014). Variants with a discrepant prediction between the programs were subsequently classified as variant of unknown significance (VUS). Variants with a concordantly predicted deleterious impact were described as candidate genes for RVOT-VT and therefore further analyzed. Variants were annotated with Alamut version 2.2 (Interactive Biosoftware).

## Two-electrode voltage-clamp (TEVC) recordings in *Xenopus laevis* oocytes

Electrophysiological studies were performed using the TEVC technique in *Xenopus laevis* oocytes. Ovarian lobes were obtained from frogs anesthetized with tricaine. Lobes were treated with collagenase (2 mg/ml, Worthington, type II) in OR2 solution (NaCl 82.5 mM, KCl 2 mM, $MgCl_2$ 1 mM, HEPES 5 mM, pH 7.4) for 120 min. Isolated oocytes were stored at 18°C in ND96 recording solution (NaCl 96 mM, KCl 2 mM, $CaCl_2$ 1.8 mM, $MgCl_2$ 1 mM, HEPES 5 mM, pH 7.5) supplemented with Na-pyruvate (275 mg/l), theophylline (90 mg/l), and gentamicin (50 mg/l). Oocytes were injected with 50 nl of cRNA for TREK-1 or TREK-1$^{I267T}$ (concentrations are indicated in each figure) and incubated for 2 days at 18°C. For the experiments analyzing the $\beta_1$-adrenoreceptor coupling, each oocyte was injected with 5 ng of TREK-1 or TREK-1$^{I267T}$ together with 20 ng of $\beta_1$-adrenoreceptor cRNA, transcribed from a pSGEM expression construct. For these experiments, oocytes were bathed after injection in a theophylline-free storage solution. Standard TEVC experiments were performed at room temperature (21–22°C) with an Axoclamp 900A amplifier, Digidata 1440A, and pClamp10 software (Axon Instruments). Microelectrodes were fabricated from glass pipettes, back-filled with 3 M KCl, and had a resistance of 0.2–1.0 MΩ.

## Animals

The investigation conforms to the guide for the Care and Use of laboratory Animals (NIH Publication 85-23). For this study, six female *Xenopus laevis* animals were used to isolate oocytes. Experiments using *Xenopus* toads were approved by the local ethics commission of the "Regierungspräsidium Giessen".

## Cell culture and transfection of HL-1 cells

HL-1 cells were grown in supplemented Claycomb medium at 37°C in an incubator supplied with 5% $CO_2$ as previously described (Claycomb *et al*, 1998). Cells were split in a 1:3 ratio when they were 100% confluent and had the ability to beat. For transfection, HL-1 cells were seeded on 35-mm dishes (Nunc) previously coated with gelatin/fibronectin in a 1:2 ratio; 24 h after seeding, cells were transfected with either 1.5 μg of TREK-1-EGFP or TREK-1$^{I267T}$-EGFP cDNA constructs using Lipofectamine 2000 (Invitrogen) following manufacturer instructions. After 24 h, transfection media was exchanged and cells were incubated for 4 h in supplemented Claycomb media. Subsequently, the 100% confluent and beating cells were used in patch clamp and imaging experiments.

## Imaging of HL-1 cells

HL-1 cells were grown on glass cover slips in 35-mm Petri dishes (Nunc), transfected as described above, and fixed with 4% PFA + 4% sucrose. Imaging was performed utilizing a Zeiss imaging platform comprising an Axio Observer.Z1 microscope equipped with a Plan-Apochromat 60×/1.40 Oil DIC objective and a standard filter set for EGFP (Zeiss 38HE). Digital images were taken with a 12 bit "AxioCam MRm" camera and processed using the AxioVision Software.

**The paper explained**

**Problem**

Approximately 60–80% of idiopathic ventricular tachycardias (VTs) arise from the right ventricle (RV), most commonly from the outflow tract (RVOT). The age of presentation is usually 30–50 years, and women are more commonly affected. Moreover, the RVOT-VTs are episodic in nature, meaning that often external triggering events are needed for the disease to become symptomatic. Here, sympathetic stimulation is the most relevant trigger. Also, due to the episodic nature of the disease, the genetic origin of RVOT-VTs is only poorly understood.

**Results**

We identified in a patient with RVOT-VTs a heterozygous point mutation in the selectivity filter of the stretch-activated $K_{2P}$ potassium channel TREK-1 (*KCNK2* or $K_{2P}2.1$). This mutation introduces abnormal sodium permeability to TREK-1. In addition, mutant channels exhibit a hypersensitivity to stretch-activation, suggesting that the selectivity filter is directly involved in stretch-induced activation and desensitization. Increased sodium permeability and stretch-sensitivity of mutant TREK-1 channels may trigger arrhythmias in areas of the heart with high physical strain such as the RVOT. In addition, while wild-type TREK-1 currents are almost fully suppressed by $\beta_1$-adrenoreceptor stimulation, the TREK-1$^{I267T}$ mutant is not responding in the typical manner and an even more pronounced inward leak current is observed. Thus, under sympathetic stimulation, the depolarizing impact of the mutant is more pronounced as under baseline conditions, an effect that might, in the patient reported, contribute to episodes of VTs under stress conditions.

**Impact**

Our findings provide important insights for future studies of $K_{2P}$ channel stretch-activation and the role of TREK-1 in mechano-electrical feedback in the heart, expanding our knowledge of the genetic basis in RVOT-VTs.

## Determination of the AP frequency in HL-1 cells

To determine the AP frequency, the beats per minute (bpm) were visually analyzed by four different persons, who simultaneously counted the beating rate in order to determine the average bpm. The average bpm was analyzed from four to six independent transfections and dishes with untransfected HL-1 cells.

## Patch clamp experiments in HL-1 cells

HL-1 cells were transfected with either TREK-1-EGFP or TREK-1$^{I267T}$-EGFP constructs as described above. APs were recorded in the whole-cell configuration under current-clamp conditions at room temperature (22°C). HL-1 cells were superfused with solution containing (in mM) 135 NaCl, 5 KCl, 1 $CaCl_2$, 1 $MgCl_2$, 0.33 $NaH_2PO_4$, 10 glucose, 2 Na-pyruvate, and 5 HEPES (pH 7.4 with NaOH) as previously described (Putzke *et al*, 2007). Pipettes pulled from borosilicate glass capillaries had a tip resistance of 3.0–4.0 MΩ when filled with a solution containing (in mM): 60 KCl, 65 K-glutamate, 5 EGTA, 2 $MgCl_2$, 3 $K_2ATP$, 0.2 $Na_2GTP$, and 5 HEPES (pH 7.2 with KOH). Data acquisition and command potentials were controlled with a commercial software program (Patchmaster, HEKA) with a sweep time interval of 1 s and a sample rate of 200 kHz. Data analysis of APs was done

    

using the Fitmaster software (HEKA). For each cell measured, the AP parameters were averaged by analyzing ten sequential APs.

## Inside-out macropatch clamp

Giant patch recordings in inside-out configuration under voltage-clamp conditions at room temperature were performed from defolliculated stage V-VI *Xenopus* oocytes 24–72 h after injection with 50 nl channel-specific cRNA. Giant patch electrodes were fabricated from thick-walled borosilicate glass. Patch pipettes were polished to give resistances of 0.3–0.7 MΩ (tip diameter of 5–15 μm) when filled with pipette solution. Standard pipette solution contained (in mM) 120 KCl, 10 HEPES, and 3.6 $CaCl_2$ (pH 7.4). The intracellular bath solution contained (in mM) 120 KCl, 10 HEPES, 2 EGTA, 1 pyrophosphate (various pHs adjusted with KOH/HCl). Data were acquired with an EPC10 amplifier (HEKA electronics, Lamprecht, Germany) and sampled at 10 kHz or higher if required and filtered with 3 kHz ($-3$ dB) or higher as appropriate for sampling rate. Macroscopic currents were recorded from different voltage pulse protocols as described for each experiment. Quantification of pressure activation currents obtained via a U-tube pressure device was compared to channel currents activated by intracellular acidification. Solutions were applied to the intracellular side of giant patches via a multi-barrel gravity flow pipette system. Tetrapentylammonium chloride (TPA-Cl) and arachidonic acid (AA) were purchased from Sigma-Aldrich and stored as stock solutions (10–100 mM) at $-80°C$ and diluted in intracellular bath solution to final concentrations prior to experimental use.

## Data analyses

Results are reported as mean $\pm$ SEM ($n$ = number of independent experiments). Every dataset for wild-type and each mutant and for every current/kinetical feature analyzed was tested with a Shapiro–Wilk test for normality. Equality of variances was tested using either a parametric or non-parametric Levene's test. In case of similar variances, significance was probed using either a paired or unpaired Student's *t*-test and for not normally distributed data, we either used a non-parametric Mann–Whitney *U*-test or for paired analyses a Wilcoxon signed-rank test. If the variances of the groups were different, significance was probed using either Welch's *t*-test, and for not normally distributed data, we used Mood's median test. *P*-values are provided within the respective Figure.

## Data availability

Data have been deposited to EGA under accession EGAS00001 002319.

**Expanded View** for this article is available online.

## Acknowledgements

This work was supported by grants of the Deutsche Forschungsgemeinschaft (DFG) (DE1482-3/2 to N.D., and Schu1082/4-2 to E.S.-B.) and by the Collaborative Research Centre SFB656 (subproject C1, to E.S.-B.). We are grateful to Michael C. Sanguinetti for providing useful comments, Oxana Nowak for excellent technical support and Erhard Wischmeyer for the β1 receptor construct.

## Author contributions

BO-B, SR, AKK, and ND performed TEVC measurements and data analyses; AKK performed experiments and data analyses with HL-1 cells; BO-B and AKK prepared or edited all figures; CF and BS analyzed WES data; CF performed QPCR experiments; MS performed inside-out macropatch clamp experiments; GS made the computational action potential modeling; RP conducted stretch experiments in HL-1 cells; ES-B, SZ, and BS enrolled subjects and contributed samples and clinical data; DB and WG performed and analyzed MD simulations; ND, ES-B, TB, JK, PK, AKK, SJ, and BS supervised the research; ND and ES-B acquired funding; ND wrote the first draft of the manuscript, and all authors contributed to the writing and editing of the revised manuscript, and approved the manuscript.

## Conflict of interest

The authors declare that they have no conflict of interest.

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
