## [Review Process File · EMBO Molecular Medicine]

Manuscript EMM-2016-06690

Sodium permeable and 'hypersensitive' TREK-1 channels cause ventricular tachycardia

Niels Decher, Beatriz Ortiz-Bonnin, Corinna Friedrich, Marcus Schewe, Aytug K Kiper, Susanne Rinné, Gunnar Seemann, Rémi Peyronnet, Sven Zumhagen, Daniel Bustos, Jens Kockskämper, Peter Kohl, Steffen Just, Wendy González, Thomas Baukrowitz, Birgit Stallmeyer, Eric Schulze-Bahr

Corresponding author: Niels Decher, Philipps-University Marburg

Review timeline:

Submission date:	09 June 2016
Editorial Decision:	25 July 2016
Revision received:	24 October 2016
Editorial Decision:	24 November 2016
Revision received:	19 January 2017
Accepted:	23 January 2017

Editor: Céline Carret

Transaction Report:

1st Editorial Decision

25 July 2016

Thank you for the submission of your manuscript to EMBO Molecular Medicine. We have now heard back from the three referees whom we asked to evaluate your manuscript. Although the referees find the study to be of interest, they also raise a number of concerns that need to be addressed in the next final version of your article.

You will see from the comments pasted below that all referees find the study interesting and novel. While referees 1 and 3 are rather enthusiastic and only have suggestions to improve clarity and conclusiveness, referee 2 is a little more demanding and would like to see more mechanism. We believe that indeed the paper would be much stronger with additional mechanistic insights to explain ventricular arrhythmia in light of the mutation but also provide the additional data to strengthen the translational aspect of the work as underlined by referee 3.

Please note that it is EMBO Molecular Medicine policy to allow only a single round of revision and that, as acceptance or rejection of the manuscript will depend on another round of review, your responses should be as complete as possible. EMBO Molecular Medicine has a "scooping protection" policy, whereby similar findings that are published by others during review or revision are not a criterion for rejection. Should you decide to submit a revised version, I do ask that you get in touch after three months if you have not completed it, to update us on the status.

Please read below for important editorial formatting.

I look forward to receiving your revised manuscript.

***** Reviewer's comments *****

Referee #1 (Remarks):

This paper by Decher et al that describes a mutation in TREK-1 as responsible for a common form of ventricular tachycardia is very interesting. This is an original finding as this channel has not yet been involved in a congenital form of cardiac arrhythmia and the functional consequence of the mutation, a loss-of-selectivity, is also unusual. The main weakness is the fact that the mutation concerns only one patient and that family members are not available for genotyping. However, one can anticipate that the publication of this mutation will trigger the research of such mutations in many other patients over the world that will confirm (or not) the involvement of TREK-1 channels in this particular type of cardiac arrhythmia.

The experiments are well designed to address most of the issues.

General comments:

The data are quite interesting but lacking clarity. This paper is hard to follow indeed. The paper may benefit from the correction by a native English speaker to improve the writing. For example, one cannot say, "TREK-1 had a linear relationship" (page 8) or "extracellular Na⁺ introduces a Na⁺ permeability and a partial reduction of outward currents in TREK-1 I267T" (page 8).

TREK-1 is highly expressed in the nervous system and in adrenal gland cortex. Surprisingly, no neurological or adrenal disorder is observed in the patient. The hypothesis proposed by the authors, i.e. an heteromerization with TREK-2 is very unlikely in the adrenal gland as TREK-2 expression in adrenal cortex is very low, more than one thousand fold lower than TREK-1 (Choi et al 2011, Science vol 331, Table S4).

As another general concern, the authors may comment on the fact that RVOT-VT arises under β -adrenergic stimulation, i.e. under elevation of the intracellular cAMP, which is a condition in which TREK-1 should be inhibited. Did the authors test a β -adrenergic stimulation in HL-1 cells overexpressing the mutant TREK-1?

Specific comments:

Fig 1E: qPCR data. It would be more interesting to compare the level of expression of TREK1 with other classic cardiac K⁺ channels such as KCNH2, KCNQ1 or KCNA5, etc...As it is, this Fig1E does not present a real interest to appreciate the importance of the TREK-1 channel in the electric activity of ventricular or atrial cardiac cell.

The {section sign} on page 7 should not be entitled "Dominant-negative Loss-of-function". The I267T TREK-1 variant produces less current than the WT, but not because of a loss-of-function. This current reduction results from the loss-of-selectivity, and with 140mM external Na⁺ ions, there is an inward Na⁺ current that develops together with the outward K⁺ current, thus making the measured outward current smaller. This cannot be called a loss-of-function. When the external Na⁺ is replaced by NMDG, the I267T TREK-1 current is not reduced anymore (fig 3D and 3F). By the way, the authors explain this on page 8.

I am not sure that chapter on the structural considerations on the binding of Na⁺ and K⁺ ions into the selectivity filter is necessary here (second {section sign} page 9, from "While K⁺ ions ... to Channel selectivity and conductivity."). It is sufficient to state that the molecular dynamic simulations are in agreement with the loss of selectivity effect of the I267T mutation, the paper is already hard enough to follow without this structural aspect.

The same comment may apply for the large discussion on the stretch-activated gating of TREK-1 (page 13-14). Such a detailed structural discussion may not be appropriate for a journal like Embo Molecular Medicine.

The authors explain the enhancement of EADs in HL1-cells overexpressing the TREK-1 I267T channel by a Ca²⁺ overload due to the Na⁺ entry triggering the reverse mode of the Na⁺/Ca²⁺ exchanger. This explanation is indeed likely, but cannot be exposed in the abstract as if it was experimentally demonstrated. Ca²⁺ imaging experiments are necessary to prove that assertion. Therefore, this explanation could be debated in the discussion section, not in the results and certainly not as evidence in the abstract.

Referee #2 (Remarks):

I reviewed the manuscript N. Decher et al. entitled "Sodium permeable and hypersensitive TREK-1 channels cause ventricular tachycardia".

In total 438 patients with various arrhythmias were screened for the presence of a KCNK2 gene mutation. In an over 40 years old patient ventricular tachycardia, emerging from the right ventricular outflow tract, without a morphological substrate was detected. Gene analysis revealed a heterozygous punctual mutation of the K2P potassium channel TREK-1.

Xenopus oocytes as well as HL-1 cells were transduced with equal amounts of TREK-1 and TREK-1I267T cRNA to mimic the heterozygous state.

Transduced oocytes showed reduced K⁺ currents and leakage for Na⁺ ions. The authors claim that a NCX mediated intracellular Ca²⁺ overload may be responsible for the observed arrhythmias.

Using TREK-1 blockers (fluoxetine, verapamil, 2-APB, riluzole, BL-1249) only BL-1249 restored channel function, most likely due to a different binding site, as authors claim.

Mechanical stress, applied via the patch clamp pipette in inside out patches, provoked higher peak currents at less negative pressure and desensitization of the stretch induced currents in TREK-1 cells was much slower than in TREK-1I267T cells.

Comments:

This is a potentially interesting manuscript. However, as it stands, the mechanisms as to how the ventricular arrhythmias are evoked remain unclear and the evidence provided is merely correlational.

- 1) Is the, for in vitro experiments used TREK-1I267T mutation identical to the TREK-1 mutation found in the patient?
- 2) It is particularly surprising that the authors are utilizing HL-1 cells as expression model, as this is an atrial tumor cell line. Instead, either hiPS cells or rodent ventricular cells need to be used as expression model to characterize the functional consequences of overexpressing the mutant ion channel.
- 3) In addition the Na-Ca exchanger and also cytosolic Ca²⁺ needs to be directly measured to underscore the working hypothesis of the proposed disease mechanism.
- 4) The stepwise applied mechanical negative pressure of 0 to -80 mmH₂O, applied by the patch clamp pipette to a very limited area of the cell membrane, is not comparable to the dynamic stress, applied to the cardiomyocytes of the RVOT during the cardiac cycle. Authors could cultivate TREK-1I267T cells on an elastic membrane, apply physiologic stress to the cellular syncytium and monitor electrophysiological characteristics.
- 5) Why the clinically observed arrhythmias become present at an age over 40 years and not in younger patients?

Referee #3 (Remarks):

Decher et al

This is a lovely submission that studies a point mutation identified in a patient with RVOT tachycardia in the TREK-1 channel. The study shows that the missense mutation alters the ion selectivity of the channel and modifies its sensitivity to mechanical stretch, two effects that would

be expected to predispose to the arrhythmia observed in the patient. Moreover, a channel activator that might prove to be a lead for therapy is identified. The manuscript is clearly written. There are a small handful of minor issues in the text to address and a two valuable bits of data would make the paper even stronger (points 6 and 8).

1. Page 7. The statement "the mutation is likely to act in a dominant-negative manner" is not well supported at this point in the paper, since decreased current could be due to an additive effect-full dominance with a dimer should decrease in current by 75% and it appears to be down only 50%. A softer assertion here could be followed by a return to the conclusion when they assess other effects that also support the notion (Em changes with substitution, effect of the activator, etc).
2. Page 9. I understand the meaning of the statement that the TWIK-like SF "is in agreement with the sodium permeability observed with TWIK-1" but it is a bit non-specific for the general reader since the change in TWIK takes many minutes to develop under pathological conditions and stated this way it sounds like it is normal TWIK operation to pass sodium.
3. Page 9. "The oxygens of the I267 carbonyl backbone coordinate..." The conclusions should be a bit less definitive since the resolution is low in the structure.
4. Page 3 or 4. Somewhere near the beginning of the paper TREK-1 should be identified with its formal IUPHAR name, K2P2, so the literature is kept clear.
5. Page 10. The nice consideration about sodium overload, calcium overload, and excess Na/Ca exchanger function is stated too much like a conclusion and is in the results. It should be noted to be a hypothesis (and maybe put in the discussion) or else intracellular Na, Ca, and exchanger function should be studied.
6. Page 11. The BL-1249 findings are really quite nice. It would be valuable to show if the activator changes the selectivity of the mutant (and if this explains the increase in current).
7. Page 11. You might cut the word "high" from before the "sensitivity to mechanical stretch" since it overstates the characteristic compared to other stretch sensitive channels and that is not necessary to make any of your strong points.
8. Page 11. It would be valuable to show if stretch changes the selectivity of the mutant (and if this explains the increase in current).
9. Citations. It seems an odd not to cite Thomas et al Neuron 2008 as proof that sodium selectivity is not just seen in the past for TWIK-1 but for TREK-1 as studied here (they showed developmental regulation by ATI produces a truncated version of TREK in the brain that can pass sodium).

1st Revision - authors' response

24 October 2016

Referee #1:

This paper by Decher et al that describes a mutation in TREK-1 as responsible for a common form of ventricular tachycardia is very interesting. This is an original finding as this channel has not yet been involved in a congenital form of cardiac arrhythmia and the functional consequence of the mutation, a loss-of-selectivity, is also unusual. The main weakness is the fact that the mutation concerns only one patient and that family members are not available for genotyping. However, one can anticipate that the publication of this mutation will trigger the research of such mutations in many other patients over the world that will confirm (or not) the involvement of TREK-1 channels in this particular type of cardiac arrhythmia. The experiments are well designed to address most of the issues.

Thank you for reviewing the manuscript and for your positive comments. Please note that we have gained within the process of the revision additional valuable insights into the disease causing mechanism of the TREK-1^{I267T} mutation. (1) RVOT-VTs occur predominantly under sympathetic stimulation. Strikingly, we found that while wild-type TREK-1 currents are almost fully suppressed by β_1 -adrenoreceptor stimulation, the TREK-1^{I267T} mutant is not responding in the typical manner, the aberrant Na⁺ conductivity persists and most importantly there are even more pronounced inward sodium leak currents with an additional depolarization of the cells. Thus, under sympathetic stimulation the depolarizing impact of the mutant is more pronounced as under baseline conditions, an effect that might, in the patient reported, contribute to episodes of VTs under stress conditions. (2) In addition, we have included more electrophysiological data using HL-1 cells, revealing that the TREK-1^{I267T} mutation strongly slows the upstroke velocity of the action potential, an effect that is known to contribute to ventricular re-entry arrhythmias due to conduction velocity reduction and thus a shorter wavelength. (3) We have included a model of the impact of the TREK-1^{I267T} mutant

on the human ventricular action potential. This data is consistent with our electrophysiological findings in HL-1 cells and predicts a secondary Ca^{2+} overload in cardiomyocytes.

General comments:

The data are quite interesting but lacking clarity. This paper is hard to follow indeed. The paper may benefit from the correction by a native English speaker to improve the writing. For example, one cannot say, "TREK-1 had a linear relationship" (page 8) or "extracellular Na^+ introduces a Na^+ permeability and a partial reduction of outward currents in TREK-1 I267T" (page 8).

Thank you for noticing these poor expressions, which were corrected in the revised manuscript. In addition, Prof. Michael Sanguinetti (Salt Lake City, University of Utah) checked the writing, to involve a native speaker, as suggested. Although referee 3 already stated that the manuscript was clearly written, we think that we have strongly improved the writing and are now much more down to the point. Please note that in the revised manuscript the corrections by the native speaker are not highlighted in gray, in order that only the changes to the content of the paper become evident. Only the changes to these two examples you mentioned are marked.

TREK-1 is highly expressed in the nervous system and in adrenal gland cortex. Surprisingly, no neurological or adrenal disorder is observed in the patient. The hypothesis proposed by the authors, i.e. an heteromerization with TREK-2 is very unlikely in the adrenal gland as TREK-2 expression in adrenal cortex is very low, more than one thousand fold lower than TREK-1 (Choi et al 2011, Science vol 331, Table S4).

We agree that the mechanism of heteromerization with TREK-2 are more likely involved in the lack of neuronal than adrenal phenotypes. In the revised Discussion section we now state:

"As the TREK-2 expression is also relatively low in the adrenal cortex (Choi et al., 2011, PMID: 21311022), other mechanisms must contribute to the lack of an adrenal phenotype, i.e. heteromerization with other K_{2P} channels expressed in the adrenal cortex like TWIK-1 (Hwang et al., 2014, PMID: 24496152; Choi et al., 2011, PMID: 21311022) or dominant-negative splice variants as TREK-1e (Rinné et al., 2014, PMID: 24196565)".

As an additional note, the expression data provided by Choi *et al.* in Supplemental Table 4 are shown in \log_2 scale and not the logarithm to the base 10 (decimal or Briggsian logarithm). Thus, with NM_021161 *KCNK10* = 5.63 and NM_001017425 *KCNK2* = 10.36 we calculate a 26.5-fold higher expression for TREK-1 ($2^{4.73}$) and not a more than thousand fold difference.

As another general concern, the authors may comment on the fact that RVOT-VT arises under β -adrenergic stimulation, i.e. under elevation of the intracellular cAMP, which is a condition in which TREK-1 should be inhibited. Did the authors test a β -adrenergic stimulation in HL-1 cells overexpressing the mutant TREK-1?

Thank you, this is an excellent comment. Please note that the original version of the manuscript already contained measurement under β -adrenergic stimulation in HL-1 cells. The changes in beating frequency observed in the original Fig. 4B were observed in the presence of 100 μM norepinephrine which is part of the original Claycomb media of HL-1 cells, presumably causing a combined activation of adrenergic α - and β -receptors which are expressed in this cell type (Filipeanu *et al.*, 2006, PMID: 16484224; McWhinney *et al.*, 2000, PMID: 11195782). The information that these recordings were performed under sympathetic stimulation is now provided in the revised Figure legend.

We performed additional experiments probing whether β_1 -receptor stimulation alters the selectivity of TREK-1^{I267T}. The new data is now included at the end of the Results section and the new Figure 7, which reads:

"Altered β_1 -adrenoreceptor modulation in TREK-1^{I267T} channels"

Attacks of RVOT-VTs arise under β -adrenergic stimulation, therefore we tested how β_1 -adrenergic stimulation influences wild-type and TREK-1^{I267T} channels co-expressed with the β_1 -adrenoreceptor in oocytes. Strikingly, while wild-type TREK-1 currents are almost fully suppressed by β_1 -adrenoreceptor stimulation (Fig 7), the TREK-1^{I267T} mutant is not responding in the typical manner (Fig 7A-C), the aberrant Na^+ conductivity persists (Fig 7B) and most importantly there are even more pronounced inward leak currents (Fig 7D and E) with an additional depolarization of the cells (Fig 7B). Thus, under sympathetic stimulation the depolarizing impact of the mutant is more pronounced as under baseline conditions, an effect that might in our particular patient contribute to episodes of VTs under stress conditions."

Specific comments:

Fig 1E: qPCR data. It would be more interesting to compare the level of expression of TREK1 with other classic cardiac K⁺ channels such as KCNH2, KCNQ1 or KCNA5, etc...As it is, this Fig1E does not present a real interest to appreciate the importance of the TREK-1 channel in the electric activity of ventricular or atrial cardiac cell.

Thank you. We agree that the data could in this form only re-confirm that there is a ventricular expression of TREK-1 in the human heart, not more. As in the initial cloning of mouse and human TREK-1 only a poor cardiac expression was reported (Fink *et al.*, 1996, PMID: 9003761; Meadows *et al.*, 2000, PMID: 10784345), these experiments were initially designed to re-confirm the ventricular TREK-1 expression. As in the meantime other groups also found a strong expression of TREK-1 in the human heart (Hund *et al.*, 2014, PMID: 24445605), the data does not provide much news value, and we have removed this data from the manuscript (Figure 1 and Results).

The {section sign} on page 7 should not be entitled "Dominant-negative Loss-of-function". The I267T TREK-1 variant produces less current than the WT, but not because of a loss-of-function. This current reduction results from the loss-of-selectivity, and with 140mM external Na⁺ ions, there is an inward Na⁺ current that develops together with the outward K⁺ current, thus making the measured outward current smaller. This cannot be called a loss-of-function. When the external Na⁺ is replaced by NMDG, the I267T TREK-1 current is not reduced anymore (fig 3D and 3F). By the way, the authors explain this on page 8.

Yes, you are right, although we mechanistically revealed at a later point in the manuscript why the TREK-1^{I267T} conducts less outward currents, we did not state this in the first electrophysiological Result section. This was done intentionally, as the experiments to proof that the apparently reduced K⁺ outward currents mechanistically result from a gain of inward Na⁺ currents, are provided only at a later point in the manuscript. Yet, we understand that the title should be changed to avoid ambiguity. We have re-named the section to “Reduced outward currents of homomeric TREK-1^{I267T} and heteromeric TREK-1/TREK-1^{I267T}”. The statement of negative dominance was also removed from the title, as suggested by Referee 3. Just a small note, the physiological external Na⁺ concentration in these oocyte experiments were 96 mM.

I am not sure that chapter on the structural considerations on the binding of Na⁺ and K⁺ ions into the selectivity filter is necessary here (second {section sign} page 9, from "While K⁺ ions ... to Channel selectivity and conductivity."). It is sufficient to state that the molecular dynamic simulations are in agreement with the loss of selectivity effect of the I267T mutation, the paper is already hard enough to follow without this structural aspect.

Thank you for this suggestion of how to improve the readability of the manuscript. We have followed your request and removed this section. Instead we now placed at the end of the preceding section “Na⁺ permeability in TREK-1^{I267T}” a statement as suggested: “In addition, we have performed Molecular Dynamic simulations which are also in agreement with the loss of selectivity effect of TREK-1^{I267T} (Appendix Fig S4).”

The same comment may apply for the large discussion on the stretch-activated gating of TREK-1 (page 13-14). Such a detailed structural discussion may not be appropriate for a journal like EMBO Molecular Medicine.

The detailed structural discussion is designed to explain the molecular mechanism for the increased stretch-sensitivity of TREK-1^{I267T} which might be a main mechanism causing the disease. Therefore, we think it belongs to the subject of Molecular Medicine. Besides the impact of the altered stretch-sensitivity of TREK-1 for this disorder, the mutant is highly relevant for the understanding of the mechanism of stretch activation in ion channels, information that we really would like to highlight.

The authors explain the enhancement of EADs in HL1-cells overexpressing the TREK-1 I267T channel by a Ca²⁺ overload due to the Na⁺ entry triggering the reverse mode of the Na⁺/Ca²⁺ exchanger. This explanation is indeed likely, but cannot be exposed in the abstract as if it was experimentally demonstrated. Ca²⁺ imaging experiments are necessary to prove that assertion. Therefore, this explanation could be debated in the discussion section, not in the results and certainly not as evidence in the abstract.

As suggested we have removed this putative explanation towards the mechanisms that is leading to the RVOT-VT arrhythmias from the Abstract and Results section. Although we did not state that HL-1 cells have an enhancement of EADs, we agree that we did not provide experiments or data to proof the idea of a secondary Ca²⁺ overload that can trigger EADs experimentally. Yet, as it is commonly accepted that an increased Na⁺ permeability is very likely to induce exactly these

secondary effects, these 'speculations' are now still, but exclusively, placed once at the end of the Discussion section.

To further support our effects observed in HL-1 cells, we have performed computational action potential modeling using a human ventricular myocyte model described by ten Tusscher (Appendix Material and Methods and Appendix Fig S6). These data were consistent with our experimental observations, and a mild depolarization and a strong slowing of the upstroke velocity was observed (Appendix Fig S6). Our action potential modeling provides also some information towards putative changes in the intracellular Ca^{2+} levels by TREK-1^{I267T}. The model was based on findings that TREK-1 in rat ventricular myocytes conducts about 1.5 pA/pF at +30 mV (Bodnár *et al.* 2015, PMID: 25539776) and our observation that TREK-1^{I267T} has a sodium permeability of 21%. According to Bodnár *et al.* when applying the electrodiffusion model developed by Goldman, Hodgkin and Katz (Hille 1992), a ventricular I_{TREK} should have at -15 mV, a potential between the E_{Na} and E_{K} , an amplitude of about 0.63 pA/pF. A respective sodium inward leak due to TREK-1^{I267T} at -15 mV was assigned with 20% of the amplitude and the opposite polarity (-0.13 pA/pF). The leak current at different potentials was subsequently calculated with a simple background current equation. Using this additional, pathological sodium background current the model predicts an increase in maximum intracellular calcium concentration of 25% (Appendix Fig S6B), maximum sarcoplasmic reticulum calcium concentration of 9% (Appendix Fig S6C), maximum intracellular sodium concentration of 17% (Appendix Fig S6D) and increased outward currents of the sodium calcium exchange current during the systole of 60.9%, reflecting an increased Ca^{2+} uptake in the reverse mode (Appendix Fig S6J). Given the limitations of a mathematical model, including that the precise magnitude of the human ventricular I_{TREK} is still unknown, we use this model only to mechanistically confirm our electrophysiological data obtained with HL-1 cells. Therefore, as described above we have limited the speculations towards a secondary calcium overload to a hypothesis which is positioned only in the last paragraph of the discussion.

Reference: B. Hille, Ionic channels of excitable membranes, Sinauer Associates, 2 edition, 1992.

Referee #2 (Remarks):

I reviewed the manuscript N. Decher et al. entitled "Sodium permeable and hypersensitive TREK-1 channels cause ventricular tachycardia". In total 438 patients with various arrhythmias were screened for the presence of a KCNK2 gene mutation. In an over 40 years old patient ventricular tachycardia, emerging from the right ventricular outflow tract, without a morphological substrate was detected. Gene analysis revealed a heterozygous punctual mutation of the K_{2P} potassium channel TREK-1. Xenopus oocytes as well as HL-1 cells were transduced with equal amounts of TREK-1 and TREK-1^{I267T} cRNA to mimic the heterozygous state. Transduced oocytes showed reduced K^+ currents and leakage for Na^+ ions. The authors claim that a NCX mediated intracellular Ca^{2+} overload may be responsible for the observed arrhythmias. Using TREK-1 blockers (fluoxetine, verapamil, 2-APB, riluzole, BL-1249) only BL-1249 restored channel function, most likely due to a different binding site, as authors claim. Mechanical stress, applied via the patch clamp pipette in inside out patches, provoked higher peak currents at less negative pressure and desensitization of the stretch induced currents in TREK-1 cells was much slower than in TREK-1^{I267T} cells.

Thank you for reviewing our manuscripts and the positive criticism. Just to be mechanistically clear, we did not transduce oocytes, they were injected with needles to ensure that a precise amount of wild-type or a mix of wild-type plus mutant cRNAs are precisely delivered to every individual oocyte. Please also note that we have gained within the process of the revision additional valuable insights into the disease causing mechanism of the TREK-1^{I267T} mutation. (1) RVOT-VTs occur predominantly under sympathetic stimulation. Strikingly, we found that while wild-type TREK-1 currents are almost fully suppressed by β_1 -adrenoreceptor stimulation, the TREK-1^{I267T} mutant is not responding in the typical manner, the aberrant Na^+ conductivity persists and most importantly there are even more pronounced inward sodium leak currents with an additional depolarization of the cells. Thus, under sympathetic stimulation the depolarizing impact of the mutant is more pronounced as under baseline conditions, an effect that might, in the patient reported, contribute to episodes of VTs under stress conditions. (2) In addition, we have included more electrophysiological data using HL-1 cells, revealing that the TREK-1^{I267T} mutation strongly slows the upstroke velocity of the action potential, an effect that is known to contribute to ventricular re-entry arrhythmias due to conduction velocity reduction and thus a shorter wavelength. (3) We have included a model of the impact of the TREK-1^{I267T} mutant on the human ventricular action potential. This data is consistent

with our electrophysiological findings in HL-1 cells and predicts a secondary Ca^{2+} overload in cardiomyocytes.

Comments:

This is a potentially interesting manuscript. However, as it stands, the mechanisms as to how the ventricular arrhythmias are evoked remain unclear and the evidence provided is merely correlational.

1) Is the, for in vitro experiments used TREK-1^{I267T} mutation identical to the TREK-1 mutation found in the patient?

Yes! It is the idea of the electrophysiological experiments to mimic exactly the mutational state and channel form present in the patient. The only exception was for the recording of the HL-1 cells which are transfected with a TREK-1 channel that carries an N-terminal EGFP-tag. For the revision we repeated the experiments in HL-1 cells with untagged TREK-1 and TREK-1^{I267T} which however gave basically the same result and thus the data were not included in the Results section of the revised manuscript.

2) It is particularly surprising that the authors are utilizing HL-1 cells as expression model, as this is an atrial tumor cell line. Instead, either hiPS cells or rodent ventricular cells need to be used as expression model to characterize the functional consequences of overexpressing the mutant ion channel.

Thank you for this suggestion, however our experience with hiPSCs was discouraging, as the cells appear to be only in a good condition, when they are in a monolayer and isolated single cells quickly seem to 'round up' and apparently go into apoptosis. Also the electrophysiological fingerprint was somewhere between an atrial and ventricular cell type (incomplete differentiation), so that we did not see the advantage for our study to use hiPSCs. We agree that using ventricular cardiomyocytes might be advantageous, however, adult ventricular cardiomyocytes are very difficult to transfect with an appropriate efficiency and thus would rather demand a viral transduction. The most suitable approach would be using adult ventricular myocytes from a transgenic TREK-1^{I267T} knock-in mouse, however the latter two approaches would represent a complete and separate story by itself. Given all these pitfalls it was the most suitable approach to use the atrial HL-1 mouse cardiomyocyte cell line. In addition, we have good experience with the electrophysiology of the HL-1 cells and can maintain and transfect them with relatively high efficiency, while they remain in the spontaneously beating stage.

3) In addition the Na-Ca exchanger and also cytosolic Ca^{2+} needs to be directly measured to underscore the working hypothesis of the proposed disease mechanism.

As suggested by Referee 1 we have removed this putative explanation towards the mechanisms that is leading to the RVOT-VT arrhythmias from the Abstract and Results section, as we agree that we did not provide experiments or data to proof the idea of a secondary Ca^{2+} overload that can trigger EADs experimentally. Yet, as it is commonly accepted that an increased Na^+ permeability is very likely to induce exactly these secondary effects, these 'speculations' are now still, but exclusively, placed once at the end of the Discussion section.

To further support our effects observed in HL-1 cells, we have performed computational action potential modeling using a human ventricular myocyte model described by ten Tusscher (Appendix Material and Methods and Appendix Fig S6). These data were consistent with our experimental observations, and a mild depolarization and a strong slowing of the upstroke velocity was observed (Appendix Fig S6). Our action potential modeling provides also some information towards putative changes in the intracellular Ca^{2+} levels by TREK-1^{I267T}. The model was based on findings that TREK-1 in rat ventricular myocytes conducts about 1.5 pA/pF at +30 mV (Bodnár *et al.* 2015, PMID: 25539776) and our observation that TREK-1^{I267T} has a sodium permeability of 21%. According to Bodnár *et al.* when applying the electrodiffusion model developed by Goldman, Hodgkin and Katz (Hille 1992), a ventricular I_{TREK} should have at -15 mV, a potential between the E_{Na} and E_{K} , an amplitude of about 0.63 pA/pF. A respective sodium inward leak due to TREK-1^{I267T} at -15 mV was assigned with 20% of the amplitude and the opposite polarity (-0.13 pA/pF). The leak current at different potentials was subsequently calculated with a simple background current equation. Using this additional, pathological sodium background current the model predicts an increase in maximum intracellular calcium concentration of 25% (Appendix Fig S6B), maximum sarcoplasmic reticulum calcium concentration of 9% (Appendix Fig S6C), maximum intracellular sodium concentration of 17% (Appendix Fig S6D) and increased outward currents of the sodium

calcium exchange current during the systole of 60.9%, reflecting an increased Ca^{2+} uptake in the reverse mode (Appendix Fig S6J). Given the limitations of a mathematical model, including that the precise magnitude of the human ventricular I_{TREK} is still unknown, we use this model only to mechanistically confirm our electrophysiological data obtained with HL-1 cells. Therefore, as described above we have limited the speculations towards a secondary calcium overload to a hypothesis which is positioned only in the last paragraph of the discussion.

Reference: B. Hille, Ionic channels of excitable membranes, Sinauer Associates, 2 edition, 1992.

4) The stepwise applied mechanical negative pressure of 0 to -80 mmH₂O, applied by the patch clamp pipette to a very limited area of the cell membrane, is not comparable to the dynamic stress, applied to the cardiomyocytes of the RVOT during the cardiac cycle. Authors could cultivate TREK-1^{I267T} cells on an elastic membrane, apply physiologic stress to the cellular syncytium and monitor electrophysiological characteristics.

We agree that it is very promising to study how a more physiological stress affects TREK-1^{I267T}. In the progress of the revision we started these experiments, trying to build up an appropriate recording rig, so we can apply cell stretch to HL-1 cells seeded on flexible silicone elastomer membranes. Here, in the three month provided for the revision, we already encountered a first major obstacle, the pre-treatment of these membranes is crucial for adhesion and proliferation of the HL-1 cells, even if all membranes were coated with the usual mix of gelatin/fibronectin. In addition, the transfection efficiency needed to be optimized under these new conditions while on top, the cells needed to keep beating. Despite some progress with the coating and transfection, we only managed to stretch HL-1 wild-type cells which results in reduced action potential frequency (video analyses). Still this new rig will require to be combined with a patch-clamp set up, in order to be able to patch the cells while stretching them. Most importantly, it seems more suitable once we have established a combined stretch/patch clamp recording rig that we also work with native ventricular myocytes as the cell shape, extracellular matrix and other factors matter for the stretch-sensitivity. Obtaining adult ventricular myocytes for which the majority or all cells carry the TREK-1^{I267T} information would demand a viral transfection or a transgenic TREK-1^{I267T} mouse line. We agree to the point you have raised, however, we strongly believe that these experiment demand a long term commitment and thus will be an independent study, which we are also aiming to target in the future.

5) Why the clinically observed arrhythmias become present at an age over 40 years and not in younger patients?

Approximately 60-80% of idiopathic VTs arise from the RV, most commonly from the outflow tract. Symptoms are related to the frequency of PVCs or non-sustained VTs. The age of presentation is usually 30-50 years (range 6-80 years) and women are more commonly affected. Often, it is likely that minor and non-sustained arrhythmias are present, but not perceived by the patient. This is also typical for other arrhythmia forms which show an age-dependence of hormonally influenced disease manifestation (e.g., Brugada syndrome or LQTS). Here, despite of having a SCN5A sodium channel mutation, the disease often remains 'hidden' (asymptomatic or neglected by the patient) until a certain age. Moreover, the diseases are episodic in nature, meaning that often external triggering events are needed in order that a disease becomes symptomatic. The reason why RVOT-VTs and certain arrhythmias occur for the first time with an age over 40 years is still poorly understood and might involve multiple factors like an age-dependent ion channel expression profile or re-modelling, fibrotic changes or even epigenetic factors that need to accumulate as a final trigger for the disease. For RVOT-VTs, the adrenergic tone is the most important trigger; so far, it is unknown whether regional RV innervation may differ between these patients.

Referee #3 (Remarks):

Decher et al

This is a lovely submission that studies a point mutation identified in a patient with RVOT tachycardia in the TREK-1 channel. The study shows that the missense mutation alters the ion selectivity of the channel and modifies its sensitivity to mechanical stretch, two effects that would be expected to predispose to the arrhythmia observed in the patient. Moreover, a channel activator that might prove to be a lead for therapy is identified. The manuscript is clearly written. There are a small handful of minor issues in the text to address and a two valuable bits of data would make the paper even stronger (points 6 and 8).

Thank you for reviewing the manuscript and for your positive comments. Please note that we have gained within the process of the revision additional valuable insights into the disease causing mechanism of the TREK-1^{I267T} mutation. (1) RVOT-VTs occur predominantly under sympathetic stimulation. Strikingly, we found that while wild-type TREK-1 currents are almost fully suppressed by β_1 -adrenoreceptor stimulation, the TREK-1^{I267T} mutant is not responding in the typical manner, the aberrant Na⁺ conductivity persists and most importantly there are even more pronounced inward sodium leak currents with an additional depolarization of the cells. Thus, under sympathetic stimulation the depolarizing impact of the mutant is more pronounced as under baseline conditions, an effect that might, in the patient reported, contribute to episodes of VTs under stress conditions. (2) In addition, we have included more electrophysiological data using HL-1 cells, revealing that the TREK-1^{I267T} mutation strongly slows the upstroke velocity of the action potential, an effect that is known to contribute to ventricular re-entry arrhythmias due to conduction velocity reduction and thus a shorter wavelength. (3) We have included a model of the impact of the TREK-1^{I267T} mutant on the human ventricular action potential. This data is consistent with our electrophysiological findings in HL-1 cells and predicts a secondary Ca²⁺ overload in cardiomyocytes.

1. Page 7. The statement "the mutation is likely to act in a dominant-negative manner" is not well supported at this point in the paper, since decreased current could be due to an additive effect-full dominance with a dimer should decrease in current by 75% and it appears to be down only 50%. A softer assertion here could be followed by a return to the conclusion when they assess other effects that also support the notion (Em changes with substitution, effect of the activator, etc).

Thank you for this comment. The lack of a perfect dominant-negative effect might also result from the problem of finding a linear relationship of 'cRNA to current' which also becomes evident for the experiments mimicking a 'haploinsufficiency'. Here with only half the amount of cRNA injected (1.25 instead of 2.5 ng) the currents were only reduced to 0.8 instead of to 0.5. Therefore, as suggested we tempered our statement to "might act" in a dominant-negative manner. In addition the title of the section was changed to: "Reduced outward currents of homomeric TREK-1^{I267T} and heteromeric TREK-1/TREK-1^{I267T}".

2. Page 9. I understand the meaning of the statement that the TWIK-like SF "is in agreement with the sodium permeability observed with TWIK-1" but it is a bit non-specific for the general reader since the change in TWIK takes many minutes to develop under pathological conditions and stated this way it sounds like it is normal TWIK operation to pass sodium.

This sentence has been re-written to avoid giving the impression that TWIK-1 is sodium permeable under baseline conditions, although we already stated at the beginning of this section that this sodium permeability occurs only under hypokalemic or acidic conditions, also providing the references by Ma *et al.*, 2011, 2012 and Chatelain *et al.* 2012. The mis-leading sentence was rephrased to "...potentially explaining the high Na⁺ ion permeability observed in the mutant channel which however already occurs under baseline conditions."

3. Page 9. "The oxygens of the I267 carbonyl backbone coordinate..." The conclusions should be a bit less definitive since the resolution is low in the structure.

As suggested by Referee 1 we have removed this section in order to improve the readability of the manuscript. Instead, as also requested by Referee 1, we now only make a brief statement towards these Molecular Dynamics simulations which can be found at the end of the preceding section "Na⁺ permeability in TREK-1^{I267T}". Here we state: "In addition, we have performed Molecular Dynamic simulations which are also in agreement with the loss of selectivity effect of TREK-1^{I267T} (Appendix Fig S4)."

4. Page 3 or 4. Somewhere near the beginning of the paper TREK-1 should be identified with its formal IUPHAR name, K2P2, so the literature is kept clear.

Good point, thank you. We now introduce in the Abstract at the first appearance of the 'historically based name' TREK-1 the IUPHAR name K_{2p}2.1 together with the gene name KCNK2. Also in the Introduction we now provide the IUPHAR name upon the first appearance of TREK-1.

5. Page 10. The nice consideration about sodium overload, calcium overload, and excess Na/Ca exchanger function is stated too much like a conclusion and is in the results. It should be noted to be a hypothesis (and maybe put in the discussion) or else intracellular Na, Ca, and exchanger function should be studied.

Thank you. As suggested by Referee 1 we have removed this putative explanation towards the mechanisms that is leading to the RVOT-VT arrhythmias from the Abstract and Results section, as we agree that we did not provide experiments or data to proof the idea of a secondary Ca^{2+} overload that can trigger EADs experimentally. Yet, as it is commonly accepted that an increased Na^+ permeability is very likely to induce exactly these secondary effects, these 'speculations' are now still, but exclusively, placed once at the end of the Discussion section.

To further support our effects observed in HL-1 cells, we have performed computational action potential modeling using a human ventricular myocyte model described by ten Tusscher (Appendix Material and Methods and Appendix Fig S6). These data were consistent with our experimental observations, and a mild depolarization and a strong slowing of the upstroke velocity was observed (Appendix Fig S6). Our action potential modeling provides also some information towards putative changes in the intracellular Ca^{2+} levels by TREK-1^{I267T}. The model was based on findings that TREK-1 in rat ventricular myocytes conducts about 1.5 pA/pF at +30 mV (Bodnár *et al.* 2015, PMID: 25539776) and our observation that TREK-1^{I267T} has a sodium permeability of 21%. According to Bodnár *et al.* when applying the electrodiffusion model developed by Goldman, Hodgkin and Katz (Hille 1992), a ventricular I_{TREK} should have at -15 mV, a potential between the E_{Na} and E_{K} , an amplitude of about 0.63 pA/pF. A respective sodium inward leak due to TREK-1^{I267T} at -15 mV was assigned with 20% of the amplitude and the opposite polarity (-0.13 pA/pF). The leak current at different potentials was subsequently calculated with a simple background current equation. Using this additional, pathological sodium background current the model predicts an increase in maximum intracellular calcium concentration of 25% (Appendix Fig S6B), maximum sarcoplasmic reticulum calcium concentration of 9% (Appendix Fig S6C), maximum intracellular sodium concentration of 17% (Appendix Fig S6D) and increased outward currents of the sodium calcium exchange current during the systole of 60.9%, reflecting an increased Ca^{2+} uptake in the reverse mode (Appendix Fig S6J). Given the limitations of a mathematical model, including that the precise magnitude of the human ventricular I_{TREK} is still unknown, we use this model only to mechanistically confirm our electrophysiological data obtained with HL-1 cells. Therefore, as described above we have limited the speculations towards a secondary calcium overload to a hypothesis which is positioned only in the last paragraph of the discussion.

Reference: B. Hille, Ionic channels of excitable membranes, Sinauer Associates, 2 edition, 1992.

6. Page 11. The BL-1249 findings are really quite nice. It would be valuable to show if the activator changes the selectivity of the mutant (and if this explains the increase in current).

Thank you. Yes, BL-1249 is able to rescue the selectivity of TREK-1^{I267T}. These data were already shown in Fig. 5E-F of the original manuscript.

7. Page 11. You might cut the word "high" from before the "sensitivity to mechanical stretch" since it overstates the characteristic compared to other stretch sensitive channels and that is not necessary to make any of your strong points.

Yes, agreed and changed.

8. Page 11. It would be valuable to show if stretch changes the selectivity of the mutant (and if this explains the increase in current).

The experiments on stretch activation were performed in symmetrical K^+ (without Na^+) and therefore a change in ion selectivity as cause for the increased currents seen for the TREK-1^{I267T} mutant upon stretch activation can be ruled out. This strongly suggests that the main effect of the mutant is, as we have proposed, caused by an altered stretch-sensitivity and stretch-desensitization and that these effects involve the selectivity filter which is the common gate for various stimuli in TREK-1 channels (Piechotta *et al.*, 2011, PMID: 21822218; Bagriantsev *et al.*, 2011, PMID: 21765396).

9. Citations. It seems an odd not to cite Thomas *et al* Neuron 2008 as proof that sodium selectivity is not just seen in the past for TWIK-1 but for TREK-1 as studied here (they showed developmental regulation by ATI produces a truncated version of TREK in the brain that can pass sodium).

Thank you we have included the following information in the Discussion: "Noteworthy, alternative translation initiation (ATI) can result in a TREK-1 variant with a truncated N-terminus (TREK-1ΔN) for which a permeability to sodium was proposed (Thomas *et al.*, 2008; PMID: 18579077). However, a recent study by Veale *et al.* leads to the conclusion that TREK-1ΔN is highly potassium selective and that the previously described effects on the reversal potentials are most likely caused by the very poor expression levels of this variant (Veale *et al.*, 2014; PMID: 24509840)."

Thank you for the submission of your revised manuscript to EMBO Molecular Medicine. We have now received the enclosed reports from the referees that were asked to re-assess it. As you will see the reviewers are now globally supportive and I am pleased to inform you that we will be able to accept your manuscript pending the following final amendments:

1) Please reply to referee 3 in a reply-to-the-referee document and amend your main text accordingly.

Please submit your revised manuscript as soon as possible.

I look forward to receiving it.

***** Reviewer's comments *****

Referee #2 (Remarks):

Most of the questions are answered. It would have been better, if authors would have performed calcium imaging experiments instead of the computational model.

Referee #3 (Comments on Novelty/Model System):

Well rationalized in the answers to review 2. I think the work stands well as it is now without transgenic animals or other methods of studying mechanical stretch.

Referee #3 (Remarks):

I remain very impressed by this study. The changes have increased the strength and clarity of the work. I have only one comment requiring attention prior to publication.

They write in the rebuttal:

9. Citations. It seems an odd not to cite Thomas et al Neuron 2008 as proof that sodium selectivity is not just seen in the past for TWIK-1 but for TREK-1 as studied here (they showed developmental regulation by ATI produces a truncated version of TREK in the brain that can pass sodium).

Thank you we have included the following information in the Discussion:

"Noteworthy, alternative translation initiation (ATI) can result in a TREK-1 variant with a truncated N-terminus (TREK-1 Δ N) for which a permeability to sodium was proposed (Thomas et al., 2008: PMID: 18579077). However, a recent study by Veale et al. leads to the conclusion that TREK-1 Δ N is highly potassium selective and that the previously described effects on the reversal potentials are most likely caused by the very poor expression levels of this variant (Veale et al., 2014; PMID: 24509840)."

However, the Veale work studied TREK-1 Δ N only in the presence of the same fenamate compounds (including BL-1249) that you show in Figure 5 restores potassium selectivity to the sodium permeable TREK variant that you study! If you want to discuss Veale in follow-up to Thomas that first reported that TREK channels can pass sodium you need to tie the results into your own exciting new findings with a sentence something like this on page 13:

"Noteworthy, alternative translation initiation (ATI) can result in a TREK-1 variant with a truncated N-terminus (TREK-1 Δ N) that is permeable to sodium (Thomas et al., 2008: PMID: 18579077). Whereas a recent study (Veale et al., 2014; PMID: 24509840) suggested that TREK-1 Δ N is highly potassium selective, they only studied the channels in the presence of fenamates (including BL-1249) to increase current density, the same agents that we show here restore potassium selectivity to the sodium permeable TREK-11267T channel (Fig 5e)."

Referee #3 (Comments on Novelty/Model System):

Well rationalized in the answers to review 2. I think the work stands well as it is now without transgenic

animals or other methods of studying mechanical stretch.

Thank you for reviewing the manuscript and for your very positive comments.

Referee #3 (Remarks):

I remain very impressed by this study. The changes have increased the strength and clarity of the work. I have only one comment requiring attention prior to publication.

....

However, the Veale work studied TREK-1ΔN only in the presence of the same fenamate compounds (including BL-1249) that you show in Figure 5 restores potassium selectivity to the sodium permeable TREK variant that you study! If you want to discuss Veale in follow-up to Thomas that first reported that TREK channels can pass sodium you need to tie the results into your own exciting new findings with a sentence something like this on page 13: "Noteworthy, alternative translation initiation (ATI) can result in a TREK-1 variant with a truncated N-terminus (TREK-1ΔN) that is permeable to sodium (Thomas et al., 2008; PMID: 18579077). Whereas a recent study (Veale et al., 2014; PMID: 24509840) suggested that TREK-1ΔN is highly potassium selective, they only studied the channels in the presence of fenamates (including BL-1249) to increase current density, the same agents that we show here restore potassium selectivity to the sodium permeable TREK-1I267T channel (Fig 5e)."

Thank you. We have included this point with almost the identical wording:

"Noteworthy, alternative translation initiation (ATI) can result in a TREK-1 variant with a truncated N-terminus (TREK-1ΔN) that is permeable to sodium (Thomas et al, 2008). While a recent study (Veale et al, 2014) suggested that TREK-1ΔN is highly potassium selective, the authors only studied the channels in the presence of fenamates (including BL-1249) to increase current density, the same agents that we show here to restore potassium selectivity in sodium permeable TREK-1^{I267T} channels (Fig 5)."

Corresponding Author Name: Niels Decher

Manuscript Number: EMM-2016-06690